# Ubiquitination of Ebola virus VP35 at lysine 309 regulates viral transcription and assembly

**Sarah van Tol**[1], **Birte Kalveram**[1], **Philipp A. Ilinykh**[2,3], **Adam Ronk**[2,3¤a], **Kai Huang**[2,3¤b¤c], **Leopoldo Aguilera-Aguirre**[1¤d], **Preeti Bharaj**[2,3¤e], **Adam Hage**[1], **Colm Atkins**[2¤f], **Maria I. Giraldo**[1], **Maki Wakamiya**[4], **Maria Gonzalez-Orozco**[1], **Abbey N. Warren**[2], **Alexander Bukreyev**[1,2,3,5], **Alexander N. Freiberg**[2,3,5,6]*, **Ricardo Rajsbaum**[1,6¤g]*

1 Department of Microbiology and Immunology, University of Texas Medical Branch, Galveston, Texas, United States of America, 2 Department of Pathology, University of Texas Medical Branch, Galveston, Texas, United States of America, 3 Galveston National Laboratory, University of Texas Medical Branch, Galveston, Texas, United States of America, 4 Transgenic Mouse Core Facility, University of Texas Medical Branch, Galveston, Texas, United States of America, 5 Center for Biodefense and Emerging Infectious Diseases, University of Texas Medical Branch, Galveston, Texas, United States of America, 6 Institute for Human Infections and Immunity, University of Texas Medical Branch, Galveston, Texas, United States of America

¤a Current address: National Security Division, Battelle Memorial Institute, Columbus, Ohio, United States of America
¤b Current address: One Health Laboratory, University of Texas Medical Branch, Galveston, Texas, United States of America
¤c Current address: Department of Internal Medicine (Infectious Diseases), University of Texas Medical Branch, Galveston, Texas, United States of America
¤d Current address: Yuva Biosciences, Birmingham, Alabama, United States of America
¤e Current address: International Center for Public Health, Rutgers University, New Jersey Medical School, Newark, New Jersey, United States of America
¤f Current address: Department of Cell Biology and Neuroscience, Rutgers University, Piscataway, New Jersey, United States of America
¤g Current address: Center for Virus-Host-Innate-Immunity, RBHS Institute for Infectious and Inflammatory Diseases, and Department of Medicine, New Jersey Medical School, Rutgers—The State University of New Jersey, Newark, New Jersey, United States of America
* anfreibe@utmb.edu (ANF); rirajsba@utmb.edu, ricardo.rajsbaum@rutgers.edu (RR)

**Data Availability Statement:** All relevant data are within the manuscript and its Supporting Information files.

## Abstract

Ebola virus (EBOV) VP35 is a polyfunctional protein involved in viral genome packaging, viral polymerase function, and host immune antagonism. The mechanisms regulating VP35's engagement in different functions are not well-understood. We previously showed that the host E3 ubiquitin ligase TRIM6 ubiquitinates VP35 at lysine 309 (K309) to facilitate virus replication. However, how K309 ubiquitination regulates the function of VP35 as the viral polymerase co-factor and the precise stage(s) of the EBOV replication cycle that require VP35 ubiquitination are not known. Here, we generated recombinant EBOVs encoding glycine (G) or arginine (R) mutations at VP35/K309 (rEBOV-VP35/K309G/-R) and show that both mutations prohibit VP35/K309 ubiquitination. The K309R mutant retains dsRNA binding and efficient type-I Interferon (IFN-I) antagonism due to the basic residue conservation. The rEBOV-VP35/K309G mutant loses the ability to efficiently antagonize the IFN-I response, while the rEBOV-VP35/K309R mutant's suppression is enhanced. The replication of both mutants was significantly attenuated in both IFN-competent and -deficient cells due to impaired interactions with the viral polymerase. The lack of ubiquitination on VP35/

**Funding:** The Rajsbaum lab is supported by NIH/NIAID (https://www.niaid.nih.gov/) grants R01 AI166668-01, R01 AI134907-01, R01 AI155466-01A1, P01 AI150585-01A1, and an UTMB IHII pilot grant (https://www.utmb.edu/ihii/programs-initiatives/pilot-grant-program), awarded to R.R., K12 HD052023 to M.I.G., T32-AI060549 and 1F31-AI15242201A1 S.v.T, and T32-AI007526 to A.H.. This work was partially supported by grant R33 AI102267 (NIH/NIAID) and a JSMEF pilot grant (https://research.utmb.edu/research-blog/research-resources-blog/2020/02/11/attention-jsmef-data-acquisition-applicants-faq) awarded to A.N.F. The funders had no role in study design, data collection and analysis, decision to publish, or preparation of the manuscript.

**Competing interests:** The authors have declared that no competing interests exist.

K309 or TRIM6 deficiency disrupts viral transcription with increasing severity along the transcriptional gradient. This disruption of the transcriptional gradient results in unbalanced viral protein production, including reduced synthesis of the viral transcription factor VP30. In addition, lack of ubiquitination on K309 results in enhanced interactions with the viral nucleoprotein and premature nucleocapsid packaging, leading to dysregulation of virus assembly. Overall, we identified a novel role of VP35 ubiquitination in coordinating viral transcription and assembly.

## Author summary

Ebola is a pathogenic zoonotic virus that causes severe human disease. Advancing our understanding of the basic molecular mechanisms underlying this virus' replication strategy is critical to expanding the availability of virus-targeted therapeutics. Here, we identified the mechanism by which EBOV VP35 ubiquitination at lysine 309 provides an advantage to complete the viral replication cycle efficiently. We utilized two virus mutants to parse the contributions of ubiquitination and a basic residue at VP35/309 to the virus life cycle. Ubiquitination is critical for facilitating optimal viral transcriptional polymerase co-factor function without affecting transcriptional initiation. The loss of a basic charge at 309 further compromises VP35's function through diminished interaction with the viral matrix protein and type-I interferon antagonism. Overall, ubiquitination and retention of a basic residue at VP35/309 is critical for viral transcription and assembly.

## Introduction

Filoviruses are a family of non-segmented, negative-sense RNA viruses that include the genus *Ebolavirus* [1]. *Zaire ebolavirus* (EBOV) has been the most devastating of the six *Ebolavirus* species to humans causing over twenty known outbreaks and nearly 15,000 deaths [2,3]. Despite the recent approval of an efficacious vaccine and development of several promising investigational antiviral and immunotherapeutic drugs [4,5], an improved understanding of EBOV's replication cycle and interaction with the host is needed to broaden the treatments to additional targets.

The EBOV genome encodes seven structural proteins: nucleoprotein (NP), polymerase co-factor (VP35), matrix protein (VP40), glycoprotein (GP), transcription factor (VP30), nucleocapsid maturation factor (VP24), and the large polymerase (L) [6,7]. Upon entry into the cell, the virus undergoes primary transcription. The EBOV transcriptase, comprised of VP35, VP30, and L, loads onto the 3' end of the genome and initiates at the transcriptional start site of the first gene, NP [6]. The transcriptase stops and re-initiates transcription at the start and stop signals along the genome template without falling off, thereby producing a gradient of viral messenger RNA (mRNA) [8–10]. Following primary transcription, the viral proteins will be in sufficient abundance for replication. The viral replicase, L with VP35, works in cooperation with NP to generate the NP-encapsidated anti-genomic RNA (cRNA) from the NP-associated genomic RNA (vRNA). Nascent vRNA is synthesized from the cRNA and serves as a template for secondary transcription or is packaged into progeny virus.

A mature nucleocapsid must be formed for EBOV vRNA to be packaged [11], and the components include the NP-vRNA, VP35, and VP24 [12]. Recruitment of VP24 results in nucleocapsid structure condensation and prevents the vRNA from acting as a template for

transcription and cRNA synthesis [13,14]. Subsequently, the mature nucleocapsid interacts with VP40 to enable budding from GP-rich membranes [11].

Regulation of the events that initiate the polymerase replicase-transcriptase transition and the formation of a mature nucleocapsid is crucial to complete the viral life cycle efficiently and to produce infectious virus. The viral and host factors that coordinate these critical stages of viral replication are unknown beyond the need for dephosphorylated VP30 for transcription [15–19] and VP24 for nucleocapsid maturation [12,13]. VP35, a component of both the active polymerase complex [6] and the mature nucleocapsid [11,12], is a potential regulator of the replicase-transcriptase transition and nucleocapsid formation. VP35 has been shown to act as an antagonist of the host's type I interferon (IFN-I) system [20–27] and a chaperone of NP to prevent premature NP homo-oligomerization and non-specific RNA encapsidation [28,29]. VP35 has also been recently found to have ATPase and helicase activities [30]. Post-translational modifications on VP35, including phosphorylation and ubiquitination, have been described to enhance VP35's polymerase co-factor activity [31–33].

We previously discovered VP35 ubiquitination at lysine (K) 309 via the host E3 ubiquitin ligase TRIM6 [31]. Ubiquitination at VP35 K309 facilitates polymerase co-factor activity. and the lack of TRIM6 impairs VP35 ubiquitination and virus replication [31]. Here, we utilized two K309 mutants, encoding either a glycine (K309G) or an arginine (K309R), to interrogate our hypothesis that VP35 ubiquitination is advantageous for virus replication. Applying these mutations in the context of recombinant viruses for infection studies and VP35 plasmids for co-expression, minigenome, and *in vitro* assays, we dissected the functional contributions of ubiquitin and a basic residue at VP35/309. Retention of a basic residue at VP35/309 is required for efficient IFN-I antagonism and interaction with multiple viral proteins, while the ubiquitination of VP35/309 promotes binding to L. Ablating the capacity for VP35/309 to receive conjugated ubiquitin impairs interaction with the viral polymerase resulting in an initiation-biased transcriptase and dysregulated intracellular viral protein proportions. The lack of ubiquitin on VP35/309 enhances VP35's interaction with NP and VP24 leading to premature nucleocapsid packaging. Based on our results, we propose that VP35/K309 ubiquitination status orchestrates VP35's engagement with the viral polymerase to facilitate the generation of a 3'-to-5' transcriptional gradient and prevents premature vRNA inaccessibility and packaging.

## Results

### A basic residue and lack of ubiquitination at VP35/309 is required for efficient IFN-I antagonism

To investigate the role(s) of VP35 K309 ubiquitination, we generated two mutants. A lysine (K) to arginine (R) mutant (K309R), intended to ablate ubiquitin conjugation without disrupting double-stranded (ds) RNA binding and dsRNA-dependent IFN-I antagonism, and a K to glycine (G) mutant (K309G), anticipated to lose both ubiquitination and IFN-I antagonism (**Fig 1A**). The substitution for a G, a small non-polar amino acid, is expected to disrupt the basic charge without disturbing the dsRNA-binding domain's structure, as has been observed for other central basic patch mutants, R312A and K339A [34]. The combined use of these mutants is intended to disentangle the importance of ubiquitination from a basic residue at position 309.

We first used purified FLAG-tagged VP35 (FLAG-VP35) to test the mutants' capacity to bind dsRNA. As predicted, the wild-type (wt) and K309R VP35 proteins were equivalent in their ability to bind dsRNA after a biotin-poly(I:C) pull-down (**Fig 1B**). In contrast, VP35-K309G binding to dsRNA was significantly decreased (approx. 50%, **Fig 1B**). This same degree of attenuation has previously been observed with a VP35 K309A mutant [35].

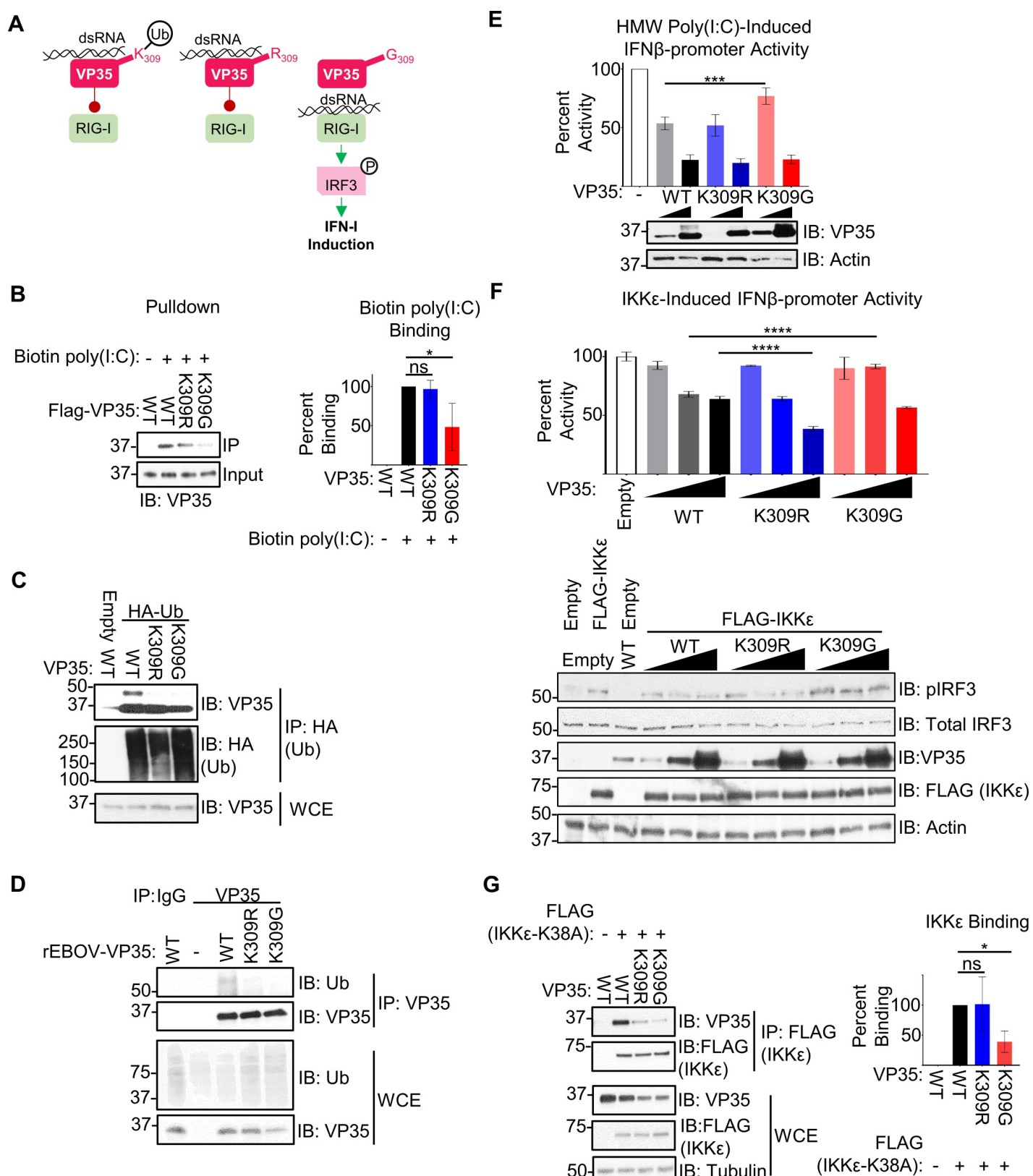

**Fig 1. A basic residue and lack of ubiquitination at VP35/309 is required for most efficient IFN-I antagonism.** (**A**) VP35 K309 is located in the IFN-inhibitory domain and is involved in binding double-stranded RNA (dsRNA) to prevent the activation of the host's cytoplasmic RNA sensor RIG-I and is ubiquitinated (white

circle with 'Ub') at this position. Mutation of K309 to an arginine (R) is predicted to prevent ubiquitination at this site without disrupting dsRNA binding due to the conservation of a basic residue. The glycine (G) mutant is predicted to lose both ubiquitination and dsRNA binding, allowing enhanced activation of RIG-I, IRF3 phosphorylation (white circle with 'P'), and downstream IFN-I induction. (**B**) Peptide purified FLAG-VP35 WT and mutants were mixed with 500 ng biotin-poly(I:C), followed by biotin pulldown. The quantification (ImageJ) represents data from three independent experiments. The percent binding was calculated as follows: the ratio of VP35 bound to poly(I:C) (IP) to the VP35 input levels for each VP35 construct was divided by the wt VP35 ratio. (**C**) Whole cell extracts (WCE) from 293T cells co-expressing HA-Ub and untagged VP35 (wt, K309R, or K309G) were used for immunoprecipitation (IP) with anti-HA beads. The presented western blot is representative of three independent experiments. (**D**) Lysates (WCE) from mock or rEBOV-VP35/wt, -K309R, or -K309G infected VeroE6 cells were used for IP with IgG (control) or anti-VP35 antibody, followed by immunoblot. The presented western blot is representative of two independent blots. (**E**) 293T cells were transfected with IFNβ luciferase reporter and Renilla luciferase plasmid and transfected 24 hours later with 3.125 ug/mL high molecular weight (HMW) poly(I:C). The ratio of firefly luciferase (IFNβ promoter activity) to renilla luciferase (transfection efficiency normalization) luminance was measured for each VP35 construct in the presence and absence of poly(I:C) stimulation. The percent activity relative to empty vector is presented. The quantification is from three independent experiments conducted in biological triplicate, and the IB is representative of the corresponding lysates. (**F**) As in E, but 2 ng IKKε was transfected along with the luciferase plasmids. The ratio of firefly luciferase (IFNβ promoter activity) to renilla luciferase (transfection efficiency normalization) luminance was measured for each VP35 construct in the presence and absence of IKKε over-expression. The percent activity relative to empty vector is presented. The quantification is from two independent experiments conducted in biological triplicate, and the IB is representative of the corresponding lysates. (**G**) Untagged VP35 constructs were incubated with FLAG-IKKε K38A, and lysates were immunoprecipitated with anti-FLAG-beads. The quantification (ImageJ) represents three independent experiments. The binding ratio ((IP: VP35/FLAG-IKKε K38A)/ (WCE: (VP35/FLAG-IKKε K38A)/Tubulin)) for each VP35 construct was divided by wt VP35's ratio to determine percent binding. Analysis was done using a one-way ANOVA with Tukey's post-test for comparison between groups. P-value: *<0.05, ***<0.001, ****<0.0001.

We then confirmed that these mutations reduce the levels of ubiquitinated VP35. In a co-immunoprecipitation assay (co-IP), after pulling down HA-tagged ubiquitin (HA-Ub) from 293T cells co-expressing VP35, the band corresponding to ubiquitin-conjugated wt VP35 was not detected for the K309R and -G mutants (**Fig 1C**). When we pulled down VP35 from infected VeroE6 cells, we observed a decrease in ubiquitin immunoprecipitated with VP35 (**Fig 1D**) which confirms that VP35/309 is ubiquitinated during infection and either K-to-R or -G mutation ablates this modification.

We then proceeded to evaluate the impact of the K309 mutations on IFN-I antagonism. In line with the impaired dsRNA binding, a low dose of VP35-K309G showed significantly less IFN antagonism as compared to wt VP35 and VP35/K309R upon stimulation with poly(I:C) in a IFNβ-promoter luciferase assay (**Fig 1E**). Other basic residues in the IFN-inhibitory domain (IID) of VP35 that are also involved in dsRNA binding [34,35] likely contribute to the VP35/K309G's antagonism at higher doses (**Fig 1E**).

VP35 also has the capacity to antagonize IFN-I induction through dsRNA binding-independent inhibition of the kinases TBK1 and IKKε [22,25], but the mechanism remains elusive. Unexpectedly, VP35/K309G was also modestly impaired in the antagonism of IKKε-induced IFNβ-luciferase promoter activity (**Fig 1F**). The VP35/K309R inhibited IFNβ induction significantly more than wt VP35 at the highest dose (**Fig 1F**), suggesting that ubiquitination on K309 may reduce the ability of VP35 to antagonize IFN production. As previously reported [25], VP35 wt binds to the IKKε kinase mutant (K38A) (**Fig 1G**). The reduced ability of VP35/K309G to interact with IKKε in a co-IP assay can explain the reduced antagonism in the luciferase assay, but IKKε-binding for the K309R mutant was not consistently affected (**Fig 1G**).

Overall, these results suggest that a substitution of either K-to-R or -G prevents ubiquitination at position 309, but only the loss of a basic residue disrupts dsRNA binding and IFN-I antagonism. In addition, lack of ubiquitination on K309 may also increase the ability of VP35 to antagonize IKKε-induced IFN production.

## Impairment of TRIM6-mediated VP35/K309 ubiquitination attenuates EBOV replication

To assess how the loss of ubiquitination at VP35/K309 affects EBOV replication, we generated recombinant EBOV (rEBOV, expressing eGFP) mutant viruses bearing the K309G or -R mutations. Since we have shown that ubiquitination on K309 promotes VP35 activity as the co-factor of the viral polymerase [31], we expected both mutants to be attenuated due to loss

of ubiquitination and that the K309G mutant would be more severely affected due to loss of full IFN antagonism activity.

IFN-competent A549 cells were infected with the recombinant viruses at a multiplicity of infection (MOI) of 0.01 plaque-forming units (PFU) per cell to evaluate multiple cycles of replication. As predicted, both of the rEBOV-VP35/K309 mutants were significantly attenuated compared to the wt virus, but the K309G showed the stronger attenuation (**Fig 2A**). This was also reflected in quantification of viral RNA expression (**Fig 2B**) and viral replication monitored by fluorescence microscopy (**Fig 2C**). Similarly, both mutant viruses were also attenuated when infecting A549 cells in a single-cycle replication kinetics experiment (MOI = 2.5 PFU/cell) (**Figs 2D, 2F,** and **S1A**). The EBOV RNA levels (**Fig 2E**), and the GFP signal (**Fig 2F**) also largely reflect these differences in titer. Attenuation of the rEBOV-VP35/K309 mutants is also observed in IFN-I competent primary murine embryonic fibroblasts (MEFs) (**S1B–S1D Figs**). Overall, at the time point corresponding to the peak titer for wt virus, the K309R mutant was attenuated 0.8 or 0.5 $\log_{10}$ and the K309G mutant was attenuated 3.1 or 2.7 $\log_{10}$ when inoculated at an MOI of 0.01 or 2.5 PFU/cell, respectively (**Fig 2G**).

We next sought to confirm that additional attenuation of rEBOV-VP35/K309G is attributable to its impaired IFN-I antagonism. The induction of IFNβ transcription (qPCR, **Figs 2H and S2A**) and secreted IFNβ (ELISA, **Fig 2I**) was higher in K309G-infected cells and lower in K309R-infected cells as compared to wt-infected cells. Induction of the IFN-stimulated genes (ISG) ISG54, Mx1, ISG15, and Ddx58 (the gene that encodes for RIG-I), showed similar dynamics to IFNβ expression with the K309G highest and the K309R lowest (**Fig 2J**). Similar patterns in IFNβ and ISG transcription were also observed during infection in primary MEFs (**S2B Fig**). Consistent with the qRT-PCR and ELISA data, activated TBK1 (pTBK1 S172), IRF3 (pIRF3 S396), and STAT1 (pSTAT1 Y701) were detected at higher levels at 24 hpi in K309G infected cells than wt-infected cells (**Fig 2K**). The K309R infected cells were depressed in TBK1, IRF3, and STAT1 activation and lagged in the induction of total STAT1 (an ISG) compared to wt-infected cells (**Fig 2K**).

To test the dependency of the attenuation on TRIM6, we infected wt or TRIM6 knockout cells [31] (T6-KO) with wt or mutant viruses. The wt virus was attenuated in the T6-KO A549 cells at low (0.01) (**Figs 3A, 3B,** and **S3A**) and high (2.5) MOI (**Figs 3C, 3D,** and **S3B**), but the K309 mutants were not additionally attenuated at peak titers. Similar results were observed during infection of primary bone marrow-derived dendritic cells (BMDCs) from newly generated *Trim6*$^{-/-}$ mice (**Fig 3E and 3F**). Early during infection, before virus could be detected in supernatants, we measured EBOV genome copy number and observed significantly less viral RNA for the wt virus in T6-KO cells but not the K309 mutants (**Fig 3E**). We then measured viral titer later during infection in supernatants from wt or *Trim6*$^{-/-}$ BMDCs infected with either the wt or K309R virus. Consistent with the results described above, the wt virus replicated significantly less in *Trim6*$^{-/-}$ cells as compared to wt BMDCs, whereas the K309R virus replicated to comparable levels in wt and *Trim6*$^{-/-}$ BMDCs (**Fig 3F**). This furthers supports a role for TRIM6 and an intact K309 residue in efficient EBOV replication.

To evaluate whether any IFN-independent factors contributed to the difference in viral load between the K309R and -G mutants, we assessed infection kinetics in IFN-incompetent cells. We expected that during infection of VeroE6, cells incapable of IFN-I production, the replication of both mutants would be attenuated compared to wt, but the difference between the mutant viruses observed in IFN-competent cells would be diminished. As predicted, both mutant viruses were attenuated compared to wt, however, the K309G mutant was more attenuated than the K309R mutant in both multi-cycle (**Fig 4A and 4B**) and single cycle (**Fig 4C and 4D**) kinetics experiments. The degree of attenuation of the K309R mutant in VeroE6 cells (0.8 and 0.5 $\log_{10}$) at the time point corresponding to the peak titer for wt virus was equivalent

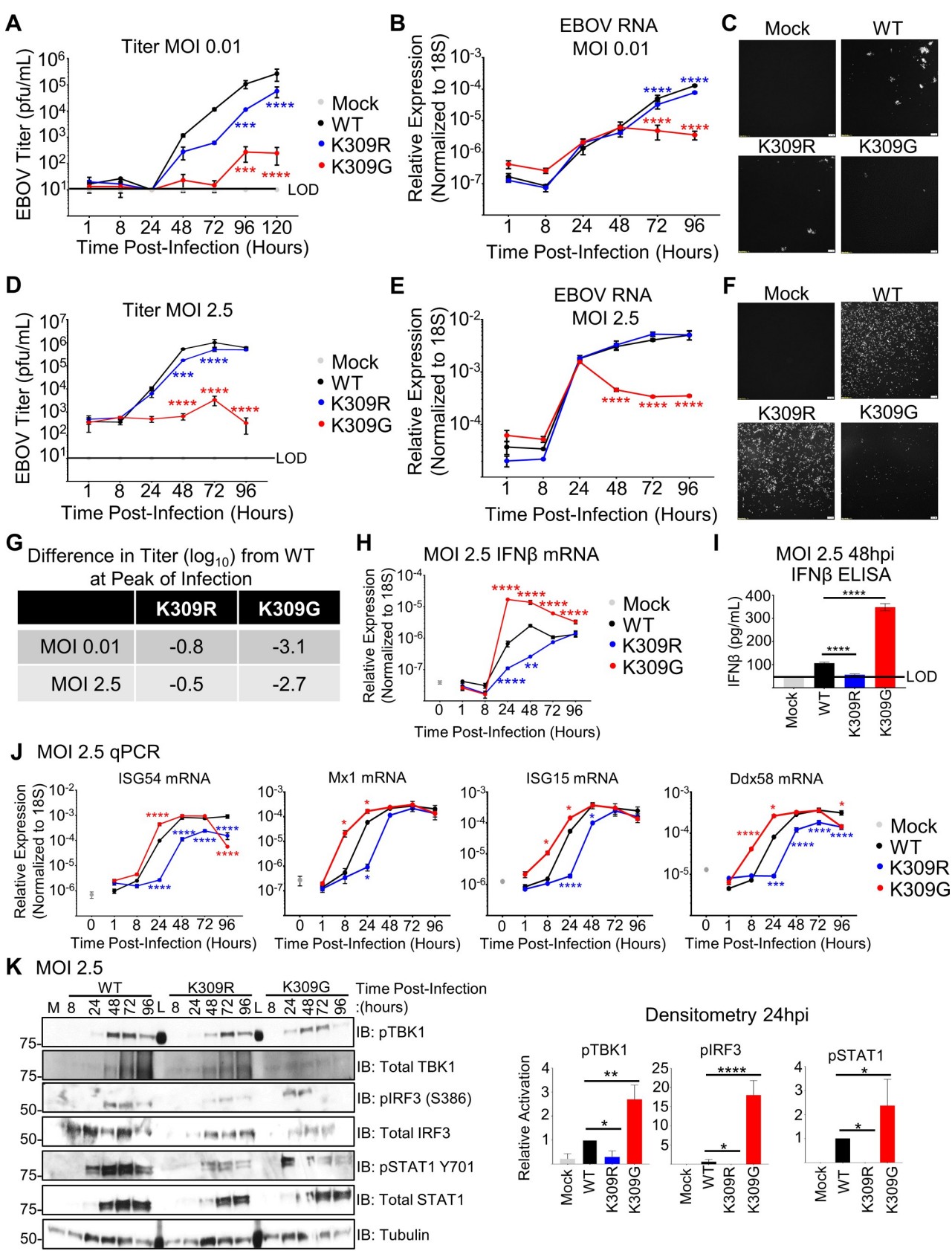

**Fig 2. The replication of rEBOV-VP35/K309R and -G mutants is attenuated in IFN-competent cells.** A549 cells were mock infected (grey) or infected in triplicate wells with rEBOV-eGFP-VP35/wt (black), -K309R (blue), or -K309G (red) at an MOI of 0.01 (**A-C**) or 2.5 PFU/cell (**D-F; J-K**). At different time points, supernatants were collected for virus titration (**A, D**) or for IFNβ ELISA (**I**, 48 hpi). The limit of detection (LOD) for the titrations (10 PFU/mL) (**A and D**) and IFNβ (50 pg/mL) (**I**) is indicated (black line). Cells were lysed in either TRIzol for RNA analysis (**B, E, H, J**) or in Laemmli buffer for immunoblot analysis (**K**). qPCR for EBOV RNA (**B and D**), IFNβ mRNA (**H**) or ISG mRNA (**J**) is shown. The fluorescence microscopy images (GFP) are representative of the three images taken (**C and F**). The difference in titer (log$_{10}$) between the mutant and wt viruses at the time point corresponding to the wt peak titer is summarized (**G**). The area under the curve (AUC) for each protein was calculated using ImageJ to determine the relative activation of the interferon pathway regulators TBK1, IRF3, and STAT1 (phosphorylated protein/(respective total protein/ tubulin)) was normalized to the activation levels in wt-infected cells. The western blots are representative of two independent experiments run in duplicate or triplicate (**K**). The titration (**A and D**), qRT-PCR (**B, D, H and J**), and ELISA (**I**) were done in biological triplicate and are representative of two independent experiments. The data analysis was done using a two-way ANOVA (**A, B, D, E, H, and J**) or one-way ANOVA (**I**) with Bonferroni's or Tukey's post-test for comparison between groups, respectively. P-value: $^*$<0.05, $^{**}$< 0.01, $^{***}$<0.001, $^{****}$<0.0001. For two-way ANOVA with Bonferroni's post-test statistical analysis, the non-significant differences (P> 0.05) are not indicated on the graph to prevent cluttering. Red and blue stars represent K309G and K309R comparison to wt, respectively.

to that observed in the A549 cells (**Figs 2G and 4E**). In contrast, the K309G mutant is less severely attenuated in VeroE6 cells (1.8 and 1.2 log$_{10}$) than in A549 cells (3.1 and 2.7 log$_{10}$) (**Figs 2G and 4E**), indicating that the attenuation of the K309G mutant virus is attributable, partially, to the reduced IFN antagonism function.

Overall, the replication kinetics experiments in the IFN-I incompetent and competent cells support that TRIM6-mediated ubiquitination of VP35/K309 promotes viral replication. The results of these experiments are also consistent with our experiments showing IFN antagonism is compromised for the K309G but bolstered for the K309R mutant. The unexpected additional attenuation of the K309G mutant in IFN-I incompetent cells suggests that a basic residue at this position is important for an additional function.

## Ubiquitination of VP35/K309 enhances viral transcriptase activity

Due to the observed attenuation of the rEBOV-VP35/K309R and -G mutant viruses in the IFN-I incompetent cells (**Fig 4A–4E**) and our previous finding that EBOV replication is impaired in TRIM6-KO cells [31], we hypothesized that ubiquitination of VP35's K309 is important for VP35's polymerase co-factor activity. To test this, we used a monocistronic firefly luciferase expressing minigenome system [6,36] co-transfected with wt or mutant VP35 plasmid. Both the VP35/K309R and -G mutants possessed equivalent polymerase co-factor activities at the lower dose (25 ng) (**Fig 5A**), used to mimic polymerase activity early during infection when the NP:VP35 ratio is higher. However, the mutants' activity was decreased approximately 50% compared to wt VP35 when using a higher dose (100 ng) (**Fig 5A**). This result supports that ubiquitination of VP35 at K309 promotes polymerase co-factor activity.

We then sought to examine the molecular mechanism by which ubiquitination of VP35 affects the viral polymerase's function. Since the luciferase readout for this minigenome system cannot differentiate between the products of viral replication and transcription, we used a strand-specific qPCR to quantify the different viral RNA species during infection [8,37]. The strand-specific strategy's use of tagged primers enables the specific transcription of viral vRNA, cRNA, or mRNAs during the reverse transcription step of cDNA synthesis (**Fig 5B**). We measured the specific viral RNA species at 48 hr in infected A549 cells, because the total viral RNA of wt and mutant viruses is similar at this time point (**Fig 2B**), allowing comparison of specific viral RNA species without bias from the viral attenuation observed at later time points. The strand-specific analysis showed that both K309 mutants had decreased levels of L mRNA (70%), and only minimal effects were observed in the vRNA and cRNA compared to wt virus (**Fig 5C**), suggesting that ubiquitination on VP35/K309 promotes efficient viral transcription.

To account for the potential effects of IFN in the attenuation of the mutant viruses and low levels of viral RNA present in the tested A549 samples, we evaluated the strand-specific RNA

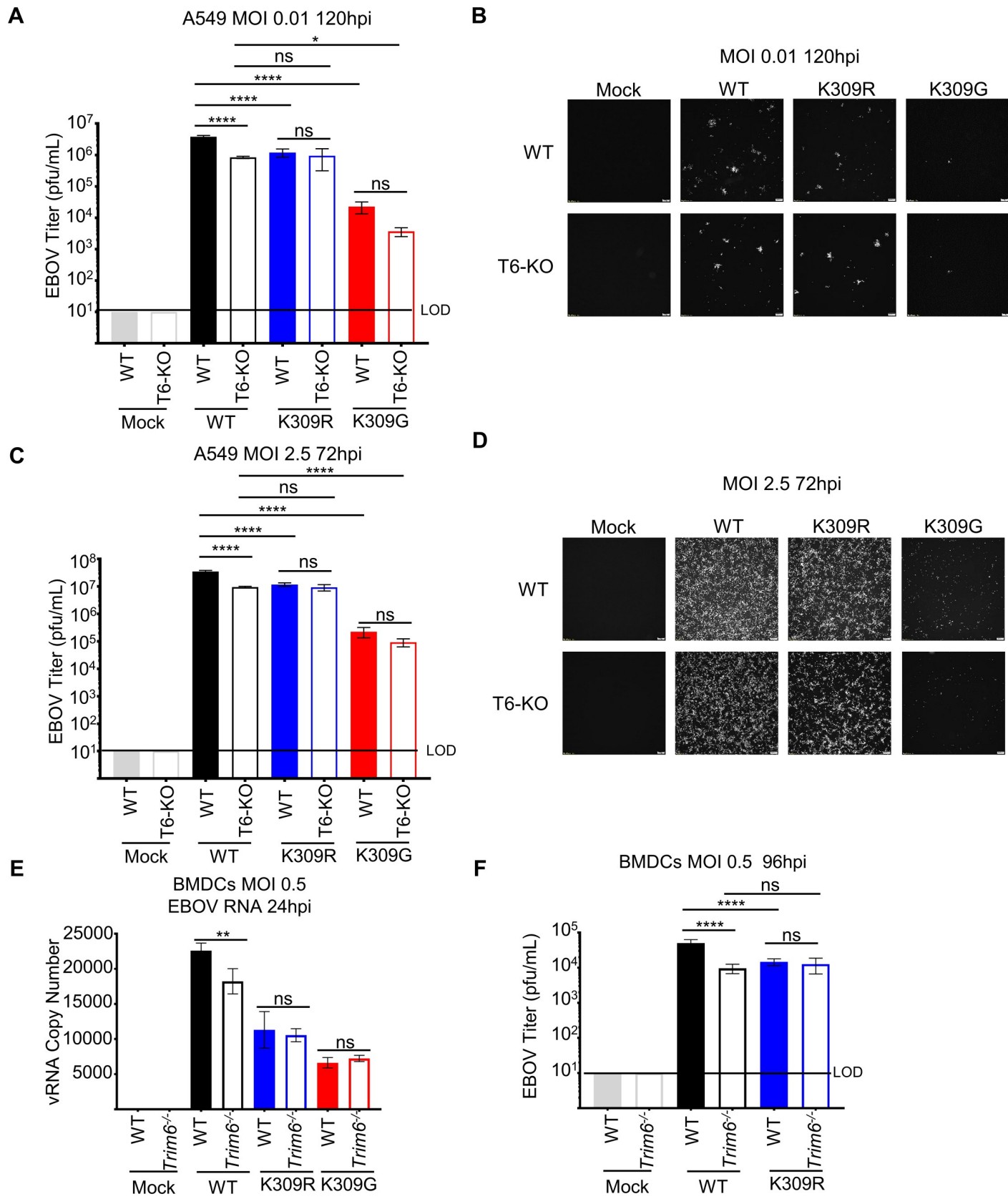

**Fig 3. TRIM6-mediated VP35/K309 ubiquitination is required for efficient replication.** Wild-type (WT) or TRIM6 knockout (T6-KO) A549 cells were infected with rEBOV-eGFP/VP35-wt, -K309R, or -K309G at a multiplicity of infection (MOI) of 0.01 PFU/cell for 120 hours (**A and B**) or 2.5 PFU/cell for 72 hours (**C and D**) corresponding to the peak titer for wt virus. The limit of detection (LOD), 10 PFU/mL, is indicated. Fluorescence images representative of three independent wells corresponding to an MOI of 0.01 PFU/cell (**B**) or 2.5 PFU/cell (**D**). (**E**) Cell-sorted CD11b⁺CD11c⁺ bone marrow-derived dendritic cells (BMDCs) from WT or *Trim6⁻/⁻* mice were infected with rEBOV-eGFP/VP35-wt, -K309R, or -K309G for 24 hours and RNA was collected for strand-specific qPCR for viral genomic RNA (vRNA). (**F**) Titer from WT or *Trim6⁻/⁻* BMDCs infected with rEBOV-eGFP/VP35-wt or -K309R at an MOI of 0.5 PFU/cell for 96 hours (n = 3). The data analysis was done using a one-way ANOVA (**A, C, E, and F**) with Tukey's post-test for comparison between groups. P-value: $^* < 0.05$, $^{**} < 0.01$, $^{****} < 0.0001$, and ns, not significant ($p > 0.05$).

production in VeroE6 cells (MOI = 1.0 PFU/cell). We also measured the NP, VP35, VP30, and VP24 mRNAs in addition to L mRNA to assess whether changes in mRNA production differ along the transcriptional gradient. Since the viral polymerase can only initiate transcription at the 3' end of the genome and must re-initiate at the next transcription start site without falling off, a 3'-to-5' transcription gradient is generated with NP and L being the most and least abundant transcripts, respectively (8–10) (**Fig 5B**). At 24 hpi, the copy number for vRNA and cRNA of the rEBOV-VP35/K309 mutants do not differ from the wt virus (**Fig 5D**). For the other EBOV genes, NP, VP35 and VP24 mRNA copy number was significantly lower for K309R as compared to wt, but not for K309G (**Fig 5D**). The copy numbers for VP40 and L transcripts were significantly lower for both mutants as compared to wt (**Fig 5D**). When looking at the ratios of copies for each RNA species relative to wt, the K309R mutant is more strongly affected than the K309G mutant and the 5' most gene, L, is the most strongly affected for both mutants (**Fig 5E**). Later in infection (72 hpi), the production of vRNA, cRNA, and all mRNAs, except NP, was lower for the K309 mutant viruses than wt (**S4A and S4B Fig**). For

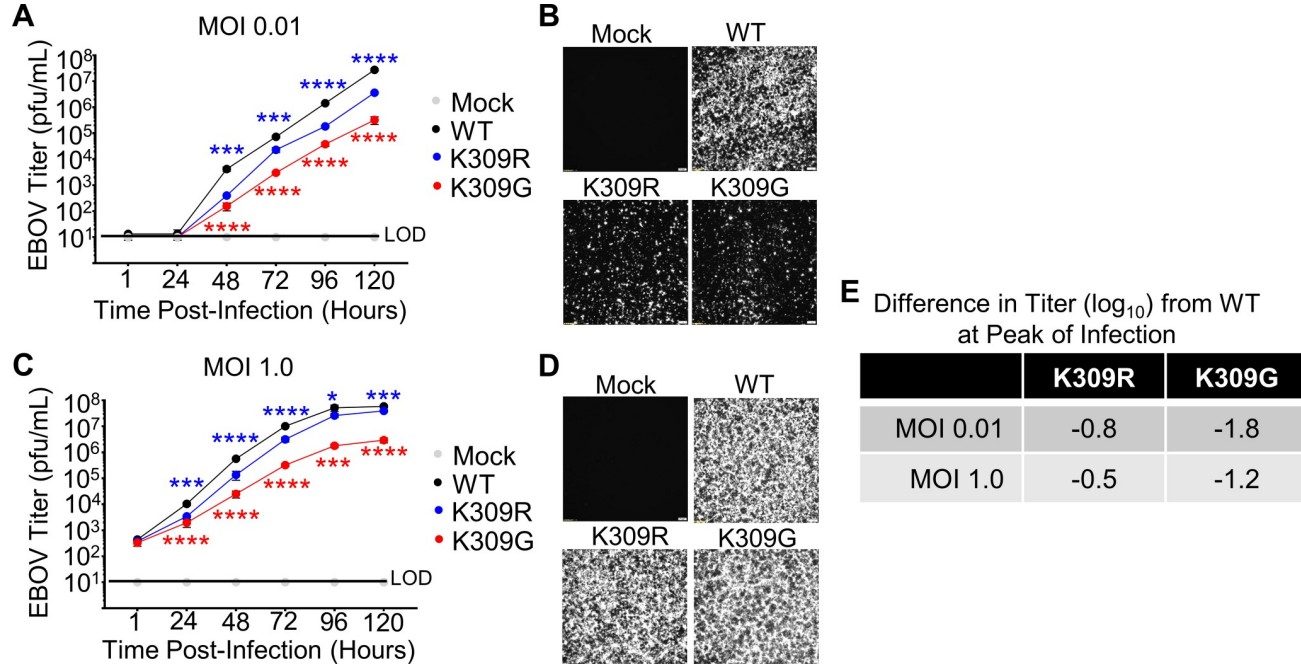

**Fig 4. The replication of rEBOV-VP35/K309R and -G mutants is attenuated in IFN-incompetent cells.** VeroE6 cells were mock infected (grey) or infected with rEBOV-eGFP-VP35/wt (black), -K309R (blue), or -K309G (red) viruses at an MOI of 0.01 (**A-B**) or 1.0 PFU/cell (**C-D**). The fluorescence microscopy images (GFP) are representative of the three images taken (**B and D**). The limit of detection (LOD), 10 PFU/mL, is indicted (**A and C**). (**E**) The difference in titer (log₁₀) between the mutant and wt viruses at the time point corresponding to the wt peak titer is summarized. The titrations were collected in biological triplicate (**A, C**). The data analysis was done using a two-way ANOVA (**A and B**) with Bonferonni's post-test for comparison between groups. P-value: $^* < 0.05$, $^{***} < 0.001$, $^{****} < 0.0001$. Red and blue stars represent K309G and K309R comparison to wt, respectively. Non-significant differences, P-value > 0.05, are not indicated to prevent cluttering on the image.

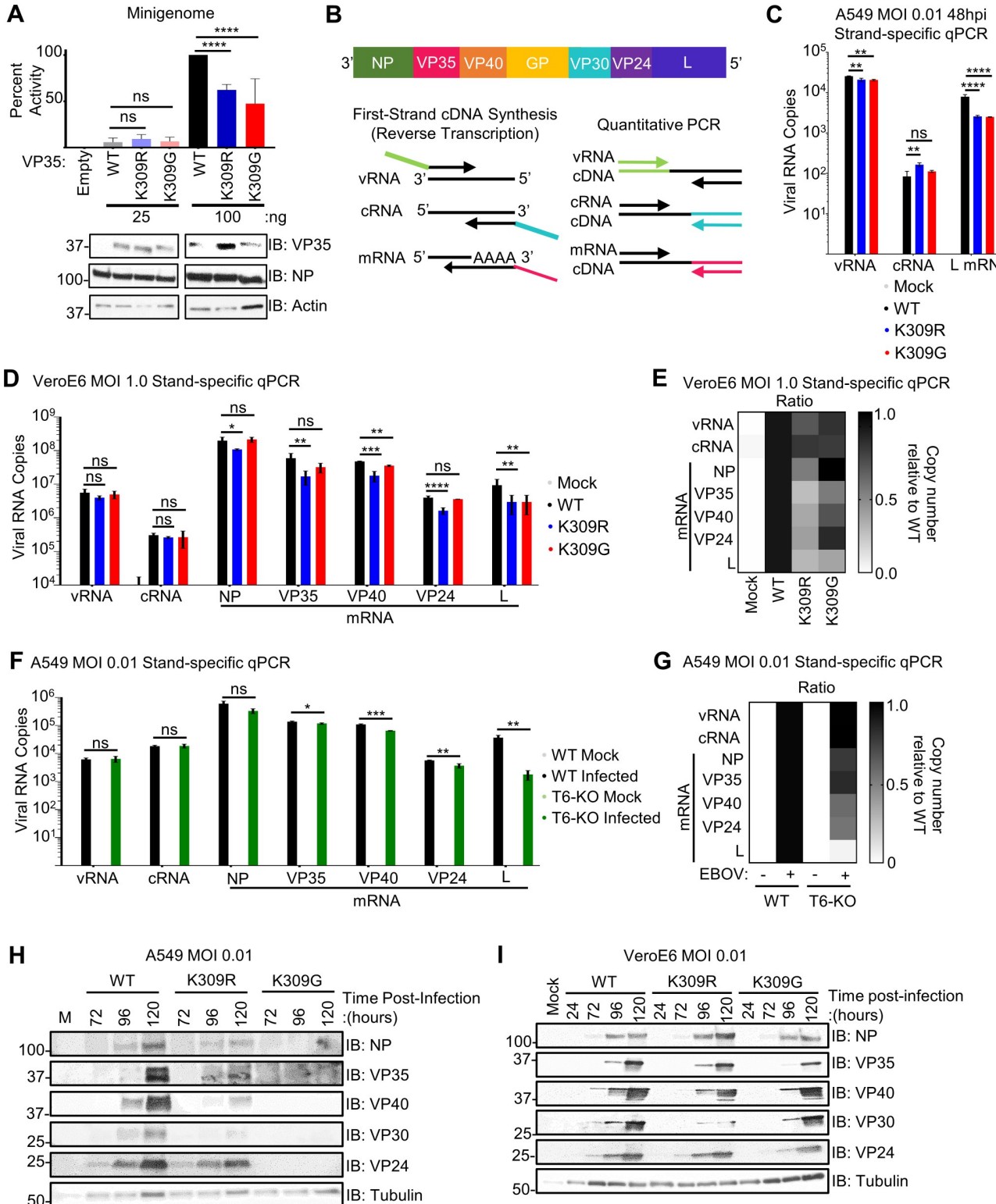

**Fig 5. Ubiquitination of VP35/309 enhances viral transcriptase function.** (**A**) Minigenome components (renilla, VP30, NP, L, T7 polymerase, and EBOV minigenome luciferase plasmid) were co-expressed with 25 or 100 ng of empty vector (pCAGGS) or VP35/wt, -K309R, or K309G in 293T cells. At 50 hours post-transfection, the cells were lysed to measure luciferase and evaluate protein expression. Quantification is from two independent experiments conducted in biological triplicate. (**B**) Graphical representation of the EBOV genome and the strand-specific qPCR approach. (**C**) A549 cells were infected with rEBOV-eGFP-VP35/wt (black), -K309R (blue), or -K309G (red) viruses at an MOI of 0.01 PFU/cell for 48 hours and RNA was

collected for strand-specific qRT-PCR (triplicates). (**D**) VeroE6 cells were mock (grey) treated or infected with rEBOV-eGFP-VP35/wt, -K309R, or -K309G viruses at an MOI of 1.0 PFU/cell for 24 hours and RNA was collected for strand-specific qRT-PCR (three biological replicates from two independent experiments with qRT-PCR run in triplicate). (**E**) Heat map representing the ratio of copy number relative to wt for each viral RNA species corresponding to the data presented in panel D. (**F**) WT (black) or TRIM6-knockout (T6-KO) (green) A549 cells were infected with rEBOV-eGFP-VP35/wt at an MOI of 2.5 PFU/cell for 24 hours and RNA was collected for strand-specific qRT-PCR (triplicates). (**G**) Heat map representing the ratio of copy number relative to wt for each viral RNA species corresponding to the data presented in panel D. (**H-I**) Protein lysates from A549 (**H**) or VeroE6 cells (**I**) infected with rEBOV-eGFP-VP35/wt, -K309R, or -K309G at an MOI of 0.01 PFU/cell, or mock-infected, were analyzed for the time-course expression of viral proteins. The data analysis was done using a one-way ANOVA with Tukey's post-test for comparison between groups (**A, C, and D**) or a student's t-test (**F**). P-value: $^{*}<0.05$, $^{**}<0.01$, $^{***}<0.001$, $^{****}<0.0001$; ns, not significant (p>0.05).

both time points, the severity of the transcriptional impairment was more pronounced for the genes on the 5' end of the genome (L > VP24, VP40, VP35 > NP) (**Figs 5D, 5E, S4A, and S4B**). With standard qPCR, which does not differentiate the viral RNA species, viral RNA did not differ among the viruses at 24 hpi but was significantly lower for both mutants as compared to wt at 72 hpi (**S4C Fig**). The increasing defect along the transcriptional gradient suggests that ubiquitination at K309 improves the polymerase's transcriptase function or stability when functioning as a transcriptase.

Since we previously reported that TRIM6 is responsible for ubiquitination on K309 [31], and TRIM6 affects viral replication in a K309-dependent manner (**Fig 3**), we assessed the synthesis of the different viral RNA species in T6-KO cells. As observed in the VeroE6 cells with the K309 mutants, the levels of vRNA, cRNA, and NP mRNA did not differ between the wt and T6-KO cells (**Fig 5F**). The significant decrease in other viral mRNAs in the T6-KO cells followed a similar pattern to the VP35/309 mutant viruses with the degree of transcriptional impairment increasing along the 3'-to-5' gradient (L > VP24, VP40 > VP35) (**Fig 5G**). Similar results were observed in primary MEFs from *Trim6*$^{-/-}$ mice infected with wt virus (**S4D and S4E Fig**).

We then evaluated whether the defect in viral transcription correlated with viral protein production. In wt and K309R-infected A549 cells, NP, VP35, and VP24 were similarly expressed, but VP40 and VP30 levels were substantially lower in K309R-infected cells (**Fig 5H**). No viral protein was detectable in K309G-infected A549 cells (**Fig 5H**). In VeroE6, as observed with the viral transcripts in the strand-specific qPCR, the viral protein expression was attenuated for both mutants with an increasing defect along the 3'-to-5' gradient, with the exception of VP24 which was affected less than VP40 (VP30 > VP40 > VP35, VP24 > NP) (**Fig 5I**). Interestingly, VP30 protein production was attenuated more strongly for the K309R mutant than for the K309G mutant (**Fig 5I**). Since the viral transcription factor is expressed measurably less by the mutants, VP30's reduced presence could perpetuate the effects of a lack of VP35 ubiquitination across multiple cycles of replication.

## Mutation of VP35/309 dysregulates VP35's interactions with the EBOV proteins but not binding with TRIM6 or itself

Due to the observed defect in polymerase co-factor activity for both VP35/309 mutants (**Fig 5A**), and because TRIM6 is able to facilitate ubiquitination of VP35 [31], we evaluated the capacity of the VP35 mutants to interact with TRIM6 and viral proteins critical for polymerase function. Co-IP experiments showed that wt and the VP35/309 mutants interact with similar efficiencies with HA-TRIM6 (**Fig 6A**). The similar binding of the mutants indicates that neither ubiquitination nor a basic residue at VP35/309 are required to interact with TRIM6.

Since oligomerization regulates multiple VP35 functions [34,38–40], we also assessed whether mutation of K309 impacts self-interaction. We co-expressed His-tagged VP35 with the corresponding FLAG-tagged VP35 construct, and the Co-IP experiment showed no obvious defect in self-interaction (**Fig 6B**).

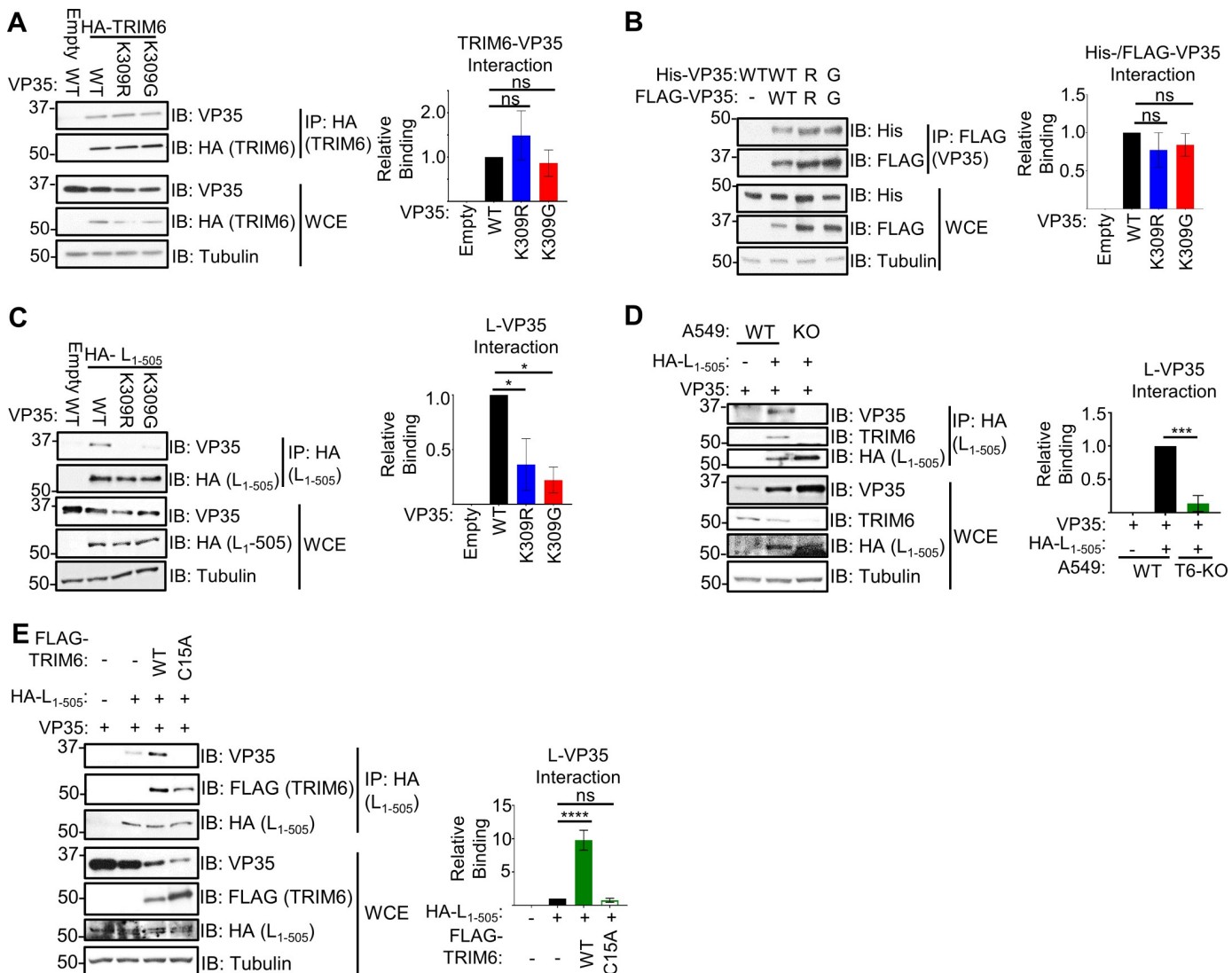

**Fig 6. Mutation of VP35 at K309 dysregulates VP35's interaction with the EBOV polymerase but not TRIM6 or itself.** (**A**) Lysates (WCE) and HA-immunoprecipitation (IP) from 293T cells co-transfected with untagged VP35 (wt, K309R, or K309G) with HA-TRIM6 or pCAGGS (empty vector). The quantification is based on immunoblot densitometry (area under the curve) determined using ImageJ from three independent experiments. The binding ratio ((IP: VP35/HA-TRIM6)/(WCE: (VP35/HA-TRIM6)/Tubulin)) for each VP35 construct was divided by wt VP35's ratio to determine relative binding. (**B**) 293T cells were co-transfected with His- or FLAG-tagged VP35 and FLAG IPs were performed. The quantification is based on AUC determined using ImageJ from three independent experiments. The binding ratio ((IP: His-VP35/FLAG-VP35)/(WCE: (His-VP35/FLAG-VP35)/Tubulin)) for each VP35 construct was divided by wt VP35's ratio to determine relative binding. (**C**) WCE and HA-IP from 293T cells co-transfected with untagged VP35 (wt, K309R, or K309G) with HA-L$_{1-505}$ or empty vector. Immunoblot quantification from two independent experiments. The binding ratio ((IP: VP35/HA-L$_{1-505}$)/(WCE: (VP35/HA-L$_{1-505}$)/Tubulin)) for each VP35 construct was divided by wt VP35's ratio to determine relative binding. (**D**) HA-L$_{1-505}$ and untagged wt VP35 were co-transfected into wt or TRIM6 knockout (T6-KO) A549 cells, and WCE were immunoprecipitated with anti-HA-tagged beads. The quantification is from data collected from three independent experiments. The binding ratio ((IP: VP35/HA-L$_{1-505}$)/(WCE: (VP35/HA-L$_{1-505}$)/Tubulin)) for lysates from wt and T6-KO cells was divided by the wt's ratio to determine relative binding. (**E**) HA-L$_{1-505}$ and untagged wt VP35 were co-transfected with FLAG-tagged TRIM6 wt or -C15A or empty vector into T6-KO A549 cells, and WCE were immunoprecipitated with anti-HA-tagged beads. The quantification is from data collected from two independent experiments. The binding ratio ((IP: VP35/HA-L$_{1-505}$)/(WCE: (VP35/HA-L$_{1-505}$)/Tubulin)) for lysates from T6-KO cells transfected with empty vector, HA-TRIM6-wt, or -C15A was divided by the ratio of empty vector transfected cells to determine relative binding. The data analysis was done using a one-way ANOVA with Tukey's post-test for comparison between groups (**A-E**). P-value: *$<0.05$, **$<0.01$, ***$<0.001$, ****$<0.0001$; ns, not significant ($p>0.05$).

To test for the interaction of VP35 with L, we used an HA-tagged N-terminal construct of L (HA-L$_{1-505}$ [41]). The N-terminus of L is sufficient to interact with VP35 [42]. Pulldown of HA-L$_{1-505}$, following co-transfection with VP35, showed impaired interaction with the VP35/

309 mutants compared to wt (**Fig 6C**). The L-VP35 interaction was disrupted to a similar degree between the K309R and -G mutants implying that ubiquitination, rather than a basic residue, is needed for high-affinity binding. TRIM6 is required for this L-VP35 interaction because ectopic expression of wt VP35 and HA-L$_{1-505}$ resulted in significantly less VP35 binding to L in T6-KO as compared to wt A549 cells (**Fig 6D**). We also observe endogenous TRIM6 being pulled down with HA-L$_{1-505}$, suggesting that VP35, L and TRIM6 may form complex (**Fig 6D**). To further test the dependence of TRIM6-mediated VP35 ubiquitination, we co-transfected T6-KO A549 cells with wt VP35 and HA-L$_{1-505}$ in the presence of wt or a catalytic TRIM6 mutant (C15A) that is unable to ligate ubiquitin onto its targets [31,43]. We observed that reconstitution with the wt, but not the C15A mutant, enhanced (10-fold) VP35's interaction with L (**Fig 6E**).

Interactions between VP35 and NP are required for coupling viral replication with NP encapsidation of the nascent vRNA or cRNA, chaperoning free NP (NP°) to prevent premature NP-RNA interaction, and assembling the nucleocapsids [28,29,41]. Further, VP35 has two distinct NP-interacting regions, one in the N-terminus [28,29] and the other in the C-terminus [41]. Due to the complexity of NP-VP35 interactions, we tested how ablation of K309 ubiquitination affects their binding in a cell-free environment. To test binding *in vitro*, lysates from FLAG-VP35 (wt, K309R, or K309G) or empty vector transfected cells were added to FLAG-beads and washed prior to the addition of purified HA-NP. Interestingly, HA-NP bound the FLAG-VP35/K309R mutant approximately 2.5-fold more efficiently than wt FLAG-VP35, but the FLAG-VP35/K309G-HA-NP interaction did not differ from wt VP35 (**Fig 7A**). This suggests that the lack of ubiquitination on VP35/309 when a basic residue is retained increases interaction with NP. To test the dependence of TRIM6 on this phenotype, we co-transfected wt VP35 and HA-NP in wt and T6-KO cells. As expected, the amount of VP35 bound to NP is higher (6-fold) in T6-KO cells (**Fig 7B**).

To test the importance of ubiquitination of VP35/309 more directly, and to rule out any minor effects due to structural changes of the K-to-R mutation, we used the ovarian tumor deubiquitinase (OTU) of Crimean-Congo hemorrhagic fever virus [44] which removes endogenous ubiquitin from modified proteins and has been used previously to demonstrate functions of ubiquitinated proteins [43,45–47]. We used the OTU to remove endogenous ubiquitin from VP35 wt prior to incubation with HA-NP-coated beads *in vitro* (**Fig 7C**). As expected, we observed more ubiquitin associated with wt- than K309R VP35 following immunoprecipitation with an anti-VP35 antibody, and the wt OTU was able to remove all the ubiquitin associated with VP35 (**Fig 7D and 7E**). VP35 was co-expressed with wt FLAG-OTU or a catalytically impaired mutant (FLAG-OTU-2A). The OTU activity was inactivated upon lysis with the deubiquitinase inhibitor N-ethylamine (NEM), and the lysates were mixed with HA-beads containing HA-NP purified from separate lysates. Consistent with the *in vitro* NP-VP35 interaction experiment, VP35 K309R binding to HA-NP was stronger (5-fold) than wt VP35 in the absence of OTU (**Fig 7D and 7F**). Co-expression with OTU-wt increased (5-fold) the amount of VP35 wt that bound HA-NP to a similar level as the untreated K309R mutant (**Fig 7D and 7F**). In contrast, co-expression with the catalytically inactive OTU-2A mutant had only minimal effects on VP35 wt's binding to HA-NP (**Fig 7D and 7F**), confirming that the lack of ubiquitin on VP35/K309 facilitates interaction with NP. The OTU co-expression did not impact the K309R mutant's interaction with NP (**Fig 7D and 7F**), suggesting that VP35/K309 ubiquitination is responsible for impeding full interaction with NP. Further, we did not observe a significant increase in the binding of a K-all-R VP35 mutant, which has all its lysine residues mutated to arginine to completely prevent ubiquitin conjugation onto VP35, to NP compared to K309R VP35 (**Fig 7D and 7F**). Overall these results support that ubiquitination onto K309 is responsible for impeding NP binding.

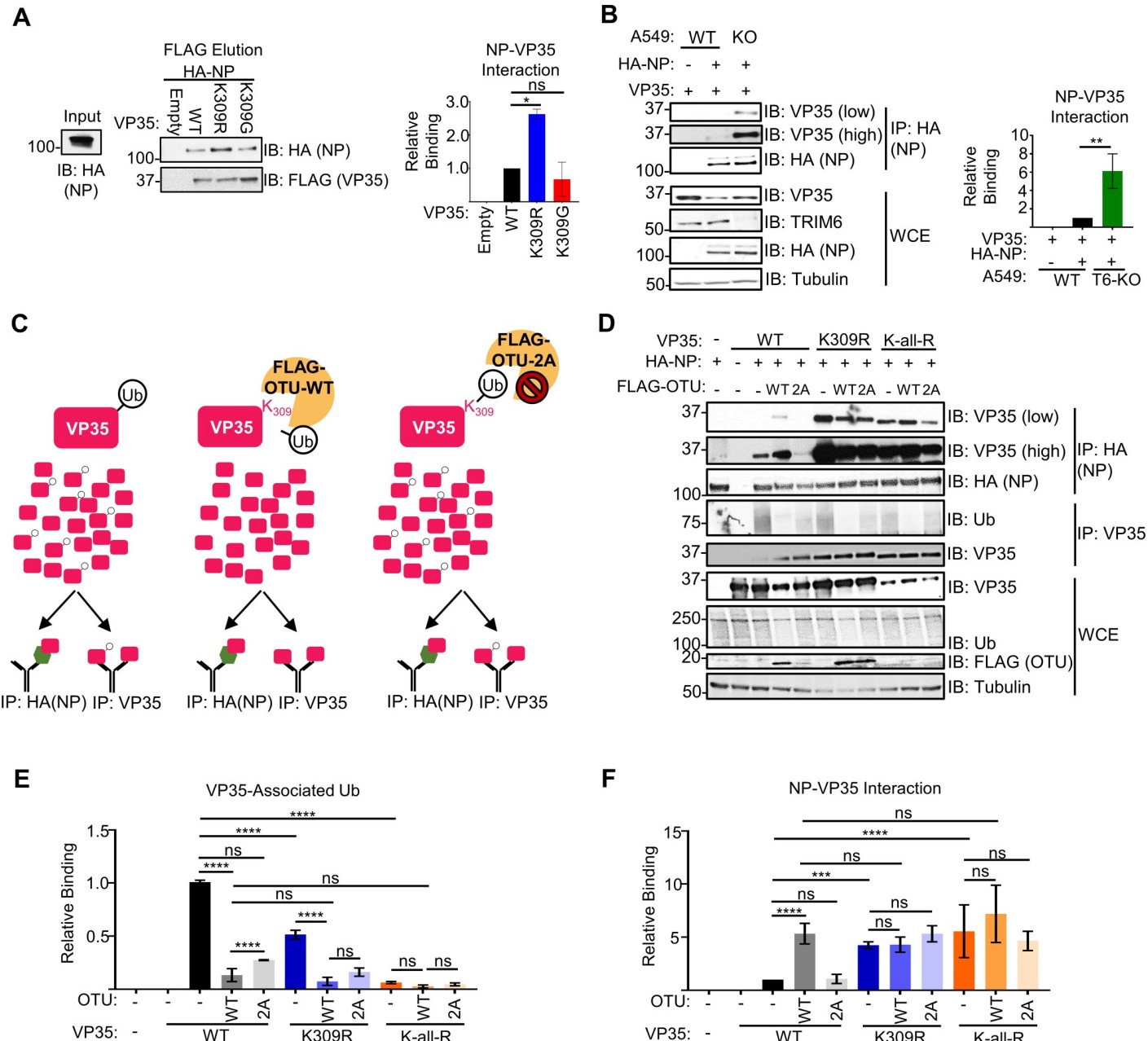

**Fig 7. Ubiquitination of VP35 at K309 impedes interaction with EBOV nucleoprotein.** (**A**) HA-NP (input) was added to beads bound with lysates from empty vector or FLAG-VP35 (wt, K309R, or K309G) transfected 293T cells, washed, and FLAG-eluted. Immunoblot quantification from two independent experiments. The binding ratio (HA-NP/FLAG-VP35) for each VP35 construct was divided by the ratio of wt VP35 to determine relative binding. (**B**) HA-NP and untagged wt VP35 were co-transfected into wt or T6-KO A549 cells and the WCE were immunoprecipitated with anti-HA beads. The blot quantifications are representative of two independent experiments. The binding ratio ((IP: VP35/HA-NP)/(WCE: (VP35/HA-NP)/Tubulin)) for lysates from wt and T6-KO cells was divided by the wt's ratio to determine relative binding. (**C**) Diagram depicting the experiment set-up for the deubiquitinase experiment. When wild-type VP35 (pink rectangle) is expressed, several ubiquitin molecules will be ubiquitinated (white circle with 'Ub') at lysine 309. When co-expressed with catalytically active, wt ovarian tumor (OTU) deubiquitinase, the covalently linked ubiquitin will be cleaved from VP35. The catalytically inactive mutant, OTU-2A, has two key cysteine residues mutated to alanine and is not able to cleave ubiquitin from substrates. Lysates cells co-expressing VP35 and FLAG-OTU were added onto beads coated with either HA-NP (antibody molecule with green hexagon) or VP35-specific antibody (antibody with pink rectangle). (**D**) 293T cells were co-transfected with untagged VP35 (wt, K309R, or K-all-R) and empty vector or FLAG-OTU (wt or -2A). The WCE from VP35 FLAG-OTU co-transfected cells were incubated with the anti-HA (IP:HA), IgG-protein A, or anti-VP35-protein (IP: VP35) coated beads, bound with lysates from HA-NP or empty vector transfected cells to pulldown VP35 (IP:HA) or ubiquitin (IP: VP35). The western blot is representative of two independent experiments run in duplicate. (**E**) The area under the curve (AUC) for each protein from the western blots in panel D were calculated using ImageJ. The relative binding ratio (VP35 IP: (Ub/VP35)/WCE: (Ub/VP35)/tubulin) for VP35-associated ubiquitin was determined for each condition and divided by the ratio for wt VP35 without OTU treatment. (**F**) The area under the curve (AUC) for each protein from the

western blots in panel D were calculated using ImageJ. The relative binding ratio (HA-NP IP: (VP35/HA-NP)/WCE: (VP35/tubulin) for VP35-NP binding was determined for each condition and divided by the ratio for wt VP35 without OTU treatment. The data analysis was done using a one-way ANOVA with Tukey's post-test (**A and B**) or two-way ANOVA with Bonferroni's post-test (**E and F**) for comparison between groups. P-value: $^*<0.05$, $^{**}<0.01$, $^{***}<0.001$, $^{****}<0.0001$; ns, not significant (p>0.05).

We also evaluated VP35's interaction with viral proteins during infection in VeroE6 cells. The K309G, but not the K309R, mutant was impaired in its interaction with NP, VP30, and VP40 compared to wt VP35 (**Fig 8A**). Both VP35/309 ubiquitination and a basic residue are important for interaction with VP24, as lack of ubiquitin increased (K309R) and loss of a basic residue (K309G) decreased this interaction (**Fig 8A**). The differential interaction between VP35 and viral proteins associated with mature nucleocapsid formation and budding, VP24 and VP40, may contribute to the excess attenuation of K309G mutant (**Fig 4A and 4C**).

## Lack of ubiquitination and a basic residue at VP35/309 dysregulates virus assembly

Due to the differences between the VP35/K309R and -G mutants in their production of infectious virus in IFN-I incompetent VeroE6 cells, despite similar levels of viral RNA, we hypothesized that both ubiquitination and retention of a basic residue at VP35/309 contribute to virus assembly. To evaluate the integrity of virus assembly, we collected samples from the supernatants and lysates of VeroE6 infected cells and the corresponding sucrose-gradient purified virus. When evaluating the relative amounts of viral proteins incorporated into the virion, the K309R and -G mutant virions contained similar amounts of NP, VP40, VP30, and VP24 relative to the amount of VP35 (**Fig 8B**). The K309G mutant trended toward lower VP35 incorporation relative to NP and VP24, but the difference was not significant (**Fig 8B**). This suggests that the K309 mutants have as similar viral protein composition to wt (**Fig 8B**) despite their dysregulated intracellular viral protein ratios (**Fig 5H and 5I**). When we measured packaged viral vRNA copies in the sucrose purified virus, the K309R mutant packaged as much vRNA as wt but the K309G mutant packaged significantly less (**Fig 8C**). As a measure of genome packaging efficiency, we compared packaged and intracellular vRNA copies and observed a 2-fold increase for the K309R mutant and a 3-fold decrease for the K309G mutant (**Fig 8D**). We then evaluated the mutants' infectivity by comparing the titer (PFU/mL) to packaged vRNA and found that the infectivity was reduced for both the K309R (78%) and -G (92%) mutants (**Fig 8E**). Finally, we compared the overall efficiency of infectious virus production by calculating the ratio of infectious virus (PFU/mL) in supernatants and the intracellular vRNA copies for the corresponding cell lysates. We found 55% ($\sim$0.4 $\log_{10}$) and 98% ($\sim$1.5 $\log_{10}$) less infectious virus/intracellular genome copy for the K309R and -G mutant viruses respectively (**Fig 8F**), which correlates with the differences observed in the kinetics experiments (**Fig 4E**). Overall, these results suggest that both ubiquitination and a basic residue at VP35/309 are important for coordinating the assembly of infectious virus. Despite the VP35/K309R mutant's increased packaging efficiency, the infectivity is impaired likely due to the premature packing of vRNA potentially associated with the increased VP24 binding. In contrast, the rEBOV-VP35/K309G mutant is impaired in both virus assembly and infectivity correlating with impaired VP35-VP24 and -VP40 interactions.

## Discussion

Ebola virus VP35 fulfills many essential roles throughout the virus' life cycle. The main functions attributed to VP35 include IFN-I antagonism and polymerase co-factor activity. VP35's role in facilitating nucleocapsid formation and genome incorporation into a virion has been

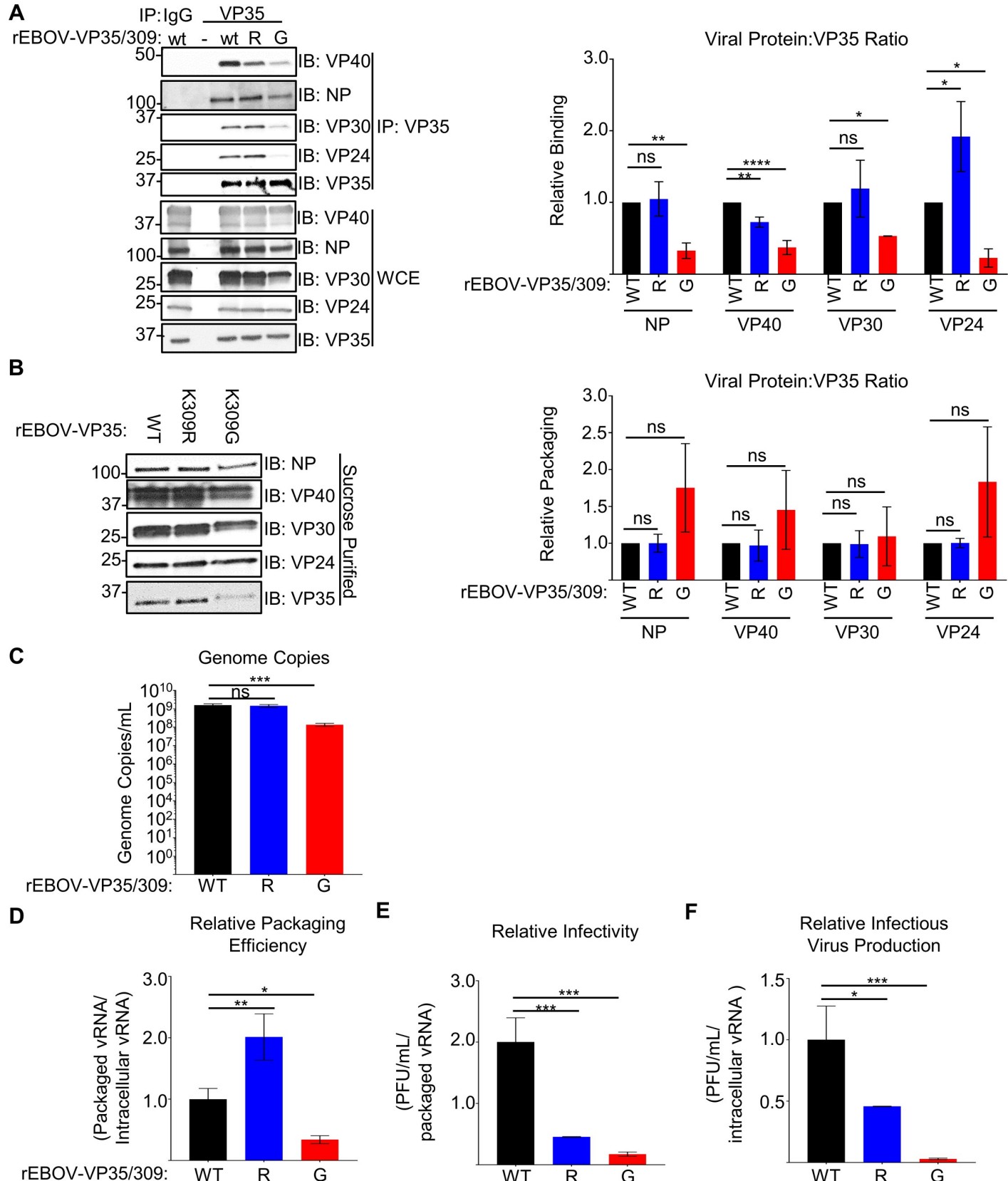

**Fig 8. Lack of ubiquitination and a basic residue at VP35/309 dysregulates virus assembly.** (**A**) Lysates from mock or rEBOV-VP35/wt/-K309R or -K309G infected (MOI = 0.01 PFU/cell for 144 hours) VeroE6 cells were immunoprecipitated (IP) with IgG or anti-VP35 antibody with protein A beads in RIPA complete and used for western blot to assess interaction with viral proteins VP40, NP, VP24 and VP30. Lysates used for this experiment were also used for Fig 1C. The area under the curve (AUC) for each protein was calculated using ImageJ from western blots run in triplicate. The relative binding ratio (IP: (viral protein/VP35)/WCE: (viral protein/VP35)) was for all VP35 constructs and divided by wt VP35's ratio. (**B**) Protein lysates (WCE) from VeroE6 cells infected cells (MOI = 0.01 PFU/cell, 144 hours) with rEBOV-eGFP-VP35/wt (WT), -K309R (R), or -K309G (G) and corresponding sucrose-gradient purified virus. The area under the curve (AUC) for each antibody were calculated using ImageJ from western blots run in triplicate. The packaging ratio (purified virus: (viral protein/VP35)/WCE (panel B): (viral protein/VP35)) was for all VP35 constructs and divided by wt VP35's ratio. (**C**) The number of viral genome copies was determined using strand-specific qPCR of sucrose-gradient-purified virus. (**D**) The ratio of packaged to intracellular RNA copies was determined using strand-specific qPCR for genomic RNA on lysates from cells and the purified virus, and the ratio was normalized to the value for wt virus. (**E**) The ratio of infectious virus to packaged genome copies was determined by titrating the sucrose-gradient purified virus (PFU/mL) and strand-specific PCR to calculate the vRNA copies in the corresponding sample, and the ratio was normalized to the value for the wt virus. (**F**) The ratio of infectious virus to intracellular genome copies was determined using the supernatant titer and intracellular genome copy number, and the ratio was normalized to the value for the wt virus. This experiment was performed in triplicate (**C-F**). The data analysis was done using a one-way ANOVA with Tukey's post-test for comparison between groups (**C-F**). P-value: $^*<0.05$, $^{**}<0.01$, $^{***}<0.001$, $^{****}<0.0001$; ns, not significant (p>0.05).

noted [12,48], but these assembly functions are not well-characterized. Further, the molecular mechanisms that govern which activity VP35 engages in have not been elucidated.

We used two different mutant recombinant viruses and VP35 expression plasmids to disassociate the importance of a basic residue from conjugated ubiquitin at residue 309. After confirming that both mutations reduced ubiquitin conjugation onto VP35, we evaluated the contribution of VP35/K309 ubiquitination in regulating VP35's dsRNA-dependent and -independent IFN antagonism activity. Based on the structure of VP35's IID and previous biochemical assays [34], VP35/K309 enhances dsRNA binding but is not required as other residues within the central basic patch directly interacting with dsRNA. This is consistent with our observations that purified FLAG-VP35/K309G binding to biotin-poly(I:C) is decreased by 50% and only the low dose of the VP35 mutant was impaired in a poly(I:C)-induced IFNβ promoter luciferase assay. Unexpectedly, we found that the K309G mutant was also deficient in IKKε binding and preventing IKKε-induced IFNβ promoter activation. The K309R mutant had enhanced dsRNA-independent IFN antagonism implying that ubiquitination on K309 reduces IFN antagonism. One potential explanation could be that ubiquitin conjugated onto VP35/K309 provides steric hindrance reducing interaction with other components of the IFN pathway. In the context of infection, the rEBOV-VP35/K309G mutant virus induced a more rapid and intense IFN-I induction than the wt virus. In contrast, IFN-I induction lags in K309R-infected cells. It is unclear whether the delay is secondary to differences in viral load or a result of VP35/K309R more efficiently antagonizing IFN-I.

Viral replication for both mutants is attenuated in IFN-I competent (A549, BMDCs, and MEFs) and -incompetent (VeroE6) cells, suggesting that preventing VP35/K309 ubiquitination impairs virus replication independent of VP35's role as an IFN-I antagonist. We previously found that TRIM6 facilitates VP35's polymerase co-factor activity using a minigenome assay [31], and we expected that blocking VP35/K309 ubiquitination directly affects VP35's co-factor function and that the attenuation is dependent on TRIM6. We elected to use A549 cells for the replication assays since we had T6-KO cells available and had characterized EBOV replication thoroughly in these cells [31], but we acknowledge that different cell types may differ in their dependence on VP35/K309 ubiquitination for efficient replication. Infection studies in T6-KO A549s, MEFs, and BMDCs demonstrate that the wt virus is attenuated when TRIM6 is absent, but the mutants' replication is not affected further. The consistent results across multiple cell types, including cells of mouse and human origin, supports that these findings are relevant. Importantly, neither mutation prevented VP35 from interacting with TRIM6. These results support that TRIM6-mediated ubiquitin conjugation to VP35/K309 affords EBOV a replication advantage.

To understand how K309 ubiquitination is advantageous for VP35's co-factor function, we measured the synthesis of different viral RNA species and evaluated VP35's interaction with the viral transcriptase. The strand-specific RNA qPCR results suggest that ubiquitination specifically benefits transcriptase but not replicase activity. Both mutants were similarly impaired in their interaction with the N-terminus of the viral polymerase (HA-L$_{1-505}$), but the K309R mutant's affinity for VP30 was not negatively impacted. The immunoprecipitation of endogenous TRIM6 with L and VP35 supports that these factors may interact during infection to promote the polymerase's transcriptase activity. The phenotypic relevance of the ubiquitin-supported interaction with L appears to present as disproportionately lower transcription of 5' viral genes relative to NP transcripts, encoded by the 3' most gene, in the rEBOV-VP35/K309R/-G-infected cells or wt virus-infected TRIM6 knockout cells. We speculate that ubiquitination may facilitate a L-VP35 interaction that favors transcriptase stability to prevent it from falling off and having to re-initiate. To our knowledge, the mechanisms regulating EBOV's transcriptase re-initiation efficiency or processivity along the genome have not yet been identified, and we provide the first evidence that ubiquitination of VP35 regulates viral transcriptase function. Alternatively, VP35 could aid the polymerase in overcoming secondary structures in the vRNA that are more abundant in the VP40, VP30, and L genes. Recently, VP35 has also been described to possess ATPase-like and helicase-like activities that are required for polymerase co-factor function [30]. Although we found the VP35/K309 mutants retain ATPase activity (S5 Fig) and recent studies suggest that residues in the N-terminus of VP35 (amino acids 137–170) are required for helicase activity [30], the potential role for ubiquitination in regulating VP35's helicase-like activity cannot be ruled out.

Removing the capacity for VP35/K309 ubiquitination results in an initiation-biased transcriptase that causes dysregulated intracellular proportions of viral proteins. The decrease in VP30 availability, particularly in the K309R-mutant infected cells, may contribute to the transcriptional defect and feedback to amplify the defect in later rounds of replication. This is in line with our observation that the transcriptional defect is more pronounced and the synthesis of vRNA and cRNA is attenuated at 72 hpi for both mutants. Reducing the intracellular pool of VP40 may translate to impaired virus assembly which is consistent with the similar amount of total viral RNA and GFP signal despite lower production of infectious virus in mutant-infected VeroE6 cells. We were unable to correlate the L protein levels due to lack of antibody, but we anticipate that lower amounts of L protein would have compounding effects on viral replication and transcription following primary transcription.

Unexpectedly, we found that the viral load in supernatants collected from rEBOV-VP35/K309G infected cells was significantly lower than the K309R mutant in IFN-I deficient cells. The difference in titer, despite similar viral RNA levels and transcriptional defects between the mutants, led us to hypothesize that a basic residue and the capacity for ubiquitin conjugation at VP35/K309 are important for virus assembly. We found that in contrast to the K309R mutant's increased interaction with VP24, the K309G-VP24 interaction is impaired. As recruitment of VP24 is important for nucleocapsid rigidification [11], the difference in the mutants' interaction with VP24 could contribute to the difference in virus production. The incorporation of VP24 to the nucleocapsid has also been reported to switch off active replication and trigger the movement of full-length/mature nucleocapsids into VP40-GP membranes [13]. As VP35/K309R interacts with VP24 more strongly than wt, the K309R mutant's lack of ubiquitination may recruit more VP24 to nucleocapsids prematurely turning off the viral polymerase activity prior to synthesizing the complete viral genome. The packaging of incomplete genomes into virions would explain the decrease in virus infectivity despite the K309R mutant's increased packaging efficiency. We also found that the VP35-VP24 interaction is impaired significantly in the K309G mutant which may prevent assembly of mature virions.

NP dynamically transitions between the NP-RNA bound and unbound states and VP24 and VP40 interact with NP [11,13,48,49]. We speculate that VP35 ubiquitination may prevent the adoption of a VP35-NP conformation that facilitates VP24 recruitment and subsequent nucleocapsid maturation.

Within EBOV-infected cells, the VP35 population includes both K309 ubiquitinated and non-ubiquitinated forms. We expect that the ability to dynamically receive and lose this post-translational modification cues VP35's interactions with other viral proteins and orchestrates VP35's engagement in its distinct functions. Although the lack of VP35/K309-ubiquitination does not prohibit the production of infectious virus, preventing this modification significantly dysregulates the viral life cycle. We propose that VP35/K309 ubiquitination facilitates a stable interaction with L to enhance viral transcription and prevents the premature packaging of immature nucleocapsids into progeny virions, and that a basic residue is needed for efficient IFN-I antagonism and interaction with VP24 (Fig 9). Overall, our results point to a novel role for host factor mediated ubiquitination in regulating Ebola virus transcription and virus assembly.

## Materials and methods

### Ethics statement

All animal procedures were conducted under animal protocols approved by the UTMB Institutional Animal Care and Use Committee and complied with USDA guidelines in an AAA-LAC-accredited lab. All experimental procedures were approved by UTMB's Institutional Biosafety Committee.

### Cells and viruses

VeroE6 cells (ATCC CRL-1586), wild-type (ATCC CCL-185) and $Trim6^{-/-}$knockout [31] A549 cells, and wt or TRIM6-knockout MEFs and BMDCs were used for infection studies. 293T cells (ATCC CRL-3216) were used for transfection. Cells were maintained in 1X DMEM (VeroE6, A549, 293T, and MEFs) or 1X RPMI-1640 (BMDCs) with 10% FBS and incubated at 37°C, 5% $CO_2$. EBOV full-length clone expressing eGFP [50] was kindly provided by Drs. Jonathan S. Towner and Stuart T. Nichol (CDC). The recombinant VP35 mutant viruses (K309R or K309G) were generated based on this clone as described [51]. Briefly, the pcDNA3 subclone containing ApaI-NruI fragment of the EBOV plasmid was subjected to mutagenesis using the Q5 site-directed mutagenesis kit (New England BioLabs). Primers used for introduction of mutations in VP35 gene are listed in S1 Table. Next, ApaI-NruI fragment in pEBOV was replaced with its mutagenized copies from pcDNA3 subclones, resulting in EBOV-eGFP/ VP35-K309R and EBOV-eGFP/VP35-K309G constructs. The recombinant viruses were recovered upon transfection of 293T cells and amplified by a single passage in VeroE6 cells. The presence of introduced mutations in viral genome was confirmed by conventional Sanger sequencing.

All manipulations with infectious EBOV were performed in the Robert E. Shope and Galveston National Laboratory Biological Safely Level 4 facilities at UTMB.

### Generation of $Trim6^{-/-}$ mice

To generate $Trim6^{-/-}$ mice using CRISPR, plasmid pSpCas9(BB)-2A-GFP [52] (gift from Feng Zhang, Addgene 48138), expressing Cas9 and sgRNA targeting exon 2 of $Trim6$ (S1 Table) was injected into the pronuclei of C57BL/6J fertilized eggs at the UTMB Transgenic Mouse Core Facility. sgRNA was designed using the following link: http://crispr.mit.edu:8079/. We

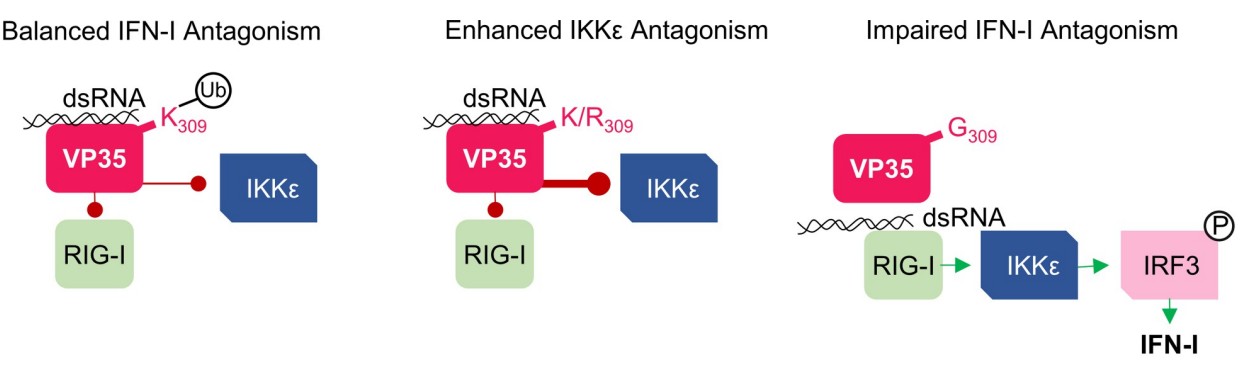

**Fig 9. A basic residue and the ubiquitination capacity of VP35/309 coordinates VP35's functions.** Both ubiquitinated (white circle with Ub) and non-ubiquitinated (K/R$_{309}$) VP35 bind double-stranded RNA (dsRNA) to antagonize RIG-I. Loss of a basic residue (G$_{309}$) impairs dsRNA binding which allows RIG-I activation and downstream IRF3 phosphorylation (white circle with P) leading to type I interferon (IFN-I) induction. In the absence of ubiquitination, VP35 (K/R$_{309}$) impedes IKKε activation more efficiently. In the context of the viral transcriptase, comprised of the viral polymerase (L), VP35, and the transcription factor (VP30), the capacity for VP35/K309 to be ubiquitinated enables balanced transcriptional activity. Under this balanced transcriptase function, a 3'-to-5' transcriptional gradient is generated and viral proteins are produced in an optimal ratio. When VP35/309 is unable to receive ubiquitination, the transcriptase is biased toward transcriptional initiation and the transcriptional gradient is dysregulated resulting in unbalanced intracellular viral protein ratios. When VP35/309 has the capacity for ubiquitination, the recruitment of VP24 and VP40 is regulated and progeny virions are assembled normally. In the absence of ubiquitin and when a basic residue is present at VP35/309, VP24 is more efficiently recruited to VP35 prematurely and some immature nucleocapsids are incorporated into progeny virions resulting in the defective viruses. When the basic residue is lost (K309G), interaction with VP24 and VP40 is impaired which reduces virus production.

used PCR (primers listed in **S1 Table**), Guide-it Mutation Detection Kit (Clontech/Takara Bio, San Jose, CA), T7E1 assay and Sanger sequencing to screen founders. Subsequently, we validated mutant 8bp deletion allele sequence (S4E **Fig**) by amplifying an exon 2 region from founder genomic DNA, subcloning the amplicons, and sequencing the amplicons. The founder line was backcrossed to C57BL/6J twice before heterozygous intercrossing. Mice were genotyped at Transnetyx (Cordova, TN).

MEFs were prepared from E14.5–15.5 embryos from wt and *Trim6*$^{-/-}$ mice and genotyped using previously described methods [53]. BMDCs were prepared as described [54]. Briefly, bone marrow cells collected from wt or T6-KO mouse femurs were incubated with 20 ng/mL GM-CSF (Biolegend) for 6 days. On the sixth day, CD11b$^+$ CD11c$^+$ cells were sorted at 98% purity (BD FACSAria Fusion–UTMB Flow Cytometry and Cell Sorting Core Lab).

Animal breeding, CRISPR-Cas9 knockout line generation, and mouse-derived primary cells preparation was performed in accordance with the approved UTMB IACUC protocols.

## Virus infections and plaque assays

Cells were plated in 10% FBS DMEM 16 hours prior to infection. The virus inoculum was prepared in 2% FBS DMEM. A portion of the inoculate was saved for back titration. At the time of infection, the medium was removed and 100 uL of the inoculum was added. The cells were incubated with the inoculum for 1 hour at 37˚C, 5% $CO_2$ and rocked every 15 minutes. The cells were washed three times with 1X DPBS (Corning) to remove the inoculum and fresh 2% FBS DMEM was added. At the indicated time points, supernatants, protein, and RNA were collected for titration, immunoblot, and qPCR respectively. An Olympus (IX73) microscope was used to take fluorescence and bright field images.

Viral titers were determined by plaque assay on Vero (CCL-81) or VeroE6 (CRL-1586) cells as previously described [31].

## Plasmids

The untagged and FLAG-tagged VP35 constructs are in the pCAGGS backbone, and together with HA-L$_{1-505}$ [41] and HA-NP were kindly provided by Dr. Basler (Mount Sinai). The K309G and K309R mutations were cloned into both the untagged and FLAG-tagged VP35 plasmids using primers (**S1 Table**) containing the appropriate point mutation and restriction enzymes sequences. To make the VP35 K-all-R mutant, we used a multistep approach to generate a VP35 construct with all sixteen lysine residues mutated to arginine (**S1 Table**). In the first step, we introduced the K6, 119, 126, and 141-to-R mutations. Then we introduced the K63,67-to-R mutations into the K6,119,126,141R construct using a two-step PCR. We then made a separate construct that introduced the K184R and K282R mutations, and this PCR product was cloned into the K6-141R mutant using the AgeI and XhoI restriction sites. The K248R and K252R mutations were inserted into the K6-141, 184,282R construct using a two-step PCR using the K309R construct as a template with the reverse primer encoding K319, 334, 339R mutations and the K6R forward primer. Finally, we mutated K216- and 222-to-R in the mutant VP35 construct using a two-step PCR. The VP35/K309R and -G constructs were amplified with primers (**S1 Table**) containing KpnI and NotI restriction sites and sub-cloned using the corresponding restriction enzymes into the His-Strep pQE TriSystem vector 1 (QIAGEN). The PCR reactions were conducted using the AccuPrime *Taq* DNA Polymerase, high fidelity kit (Invitrogen). The mutant plasmids sequences were confirmed using Sanger sequencing (UTMB Molecular Genomics).

The other plasmids including HA-Ub, HA-TRIM6, Renilla luciferase, IFNβ-luciferase promoter, FLAG-IKKε (wt and K38A), pCAGGS empty vector, FLAG-OTU-wt, FLAG-OTU-2A

[43] and minigenome components (EBOV L, EBOV NP, EBOV VP30, T7 polymerase, and EBOV minigenome firefly luciferase) [31] have been previously described. Using HA-TRIM6 wt or C15A pCAGGS plasmids as a template, we PCR amplified these constructs with primers containing restriction sites SgfI and MluI to sub-clone the products into pCMV6-FLAG-Myc vector (**S1 Table**).

## Transfections and Immunoprecipitations

293Ts were plated in 6-well plates (400,000 cells/well) in 10% FBS DMEM for 16 hours, followed by transfection using *Trans*IT-LT1 (Mirus) following the manufacturer's recommendations. A549 wt or T6-KO cells we plated in 6-well plates (400,000 cells/well) in 10% FBS DMEM for 16 hours followed by transfection using Lipofectamine 3000 (Thermo Scientific) and the media was changed 5–6 hours after transfection. Twenty-eight hours after transfection, 293T or A549 cells were lysed in RIPA buffer with complete protease inhibitor (Roche), n-ethylmaleimide (NEM), and iodoacetamide (IA) (RIPA complete). Lysates were cleared at 25,200 g for 20 minutes at 4°C, and 10% of the clarified lysate was added to 2X Laemmli sample buffer (BioRad) with 5% beta-mercaptoethanol and boiled at 95°C for 10 minutes to generate whole cell extracts (WCE). The remaining clarified lysate was mixed with 7.5 uL of anti-HA-Agarose beads (Sigma) or anti-FLAG-Agarose beads (Sigma) and incubated at 4°C overnight on a rotating platform. For co-IP from infected cells, VeroE6 cells infected at an MOI of 0.01 PFU/cell were lysed in RIPA complete at 144 hpi. The clarified lysates were incubated with 1 ug of anti-mouse-IgG (BD Biosciences) or -VP35 (Kerafast) antibody and protein A beads (Cytiva) overnight. The beads were washed seven times with RIPA buffer with IA and NEM before boiling in 2X Laemmli buffer (HA and FLAG co-IP) or 65 uL RIPA complete with 25 uL 4X NuPAGE LDS Sample buffer (Thermo Scientific) and 10 uL 10X NuPAGE Sample Reducing Reagent (Thermo Scientific) (VP35/IgG co-IP).

## Protein purification

To collect purified HA- or FLAG-tagged proteins, we transfected 293T cells and immunoprecipitated with anti-HA or -FLAG beads as described above prior to peptide elution. After the seven washes in 1X TBS-T, beads were washed once in peptide elution buffer (10 mM TRIS pH 7.4 and 150 mM NaCl in nuclease free water (NF $H_2O$)) without peptide. The protein was then eluted in 15 μL of peptide elution buffer with HA- (1 mg/mL) or FLAG- (300 μg/mL) peptide three times. The peptide purified protein was aliquoted and stored at -80°C until use.

## IFNβ luciferase promoter assays

293T cells were plated in a 96-well plate (20,000 cells/well) in 10% FBS DMEM for 16 hours prior to transfection. For the IKKε-induction experiment, cells were co-transfected with 20 ng renilla (to normalize transfection efficiency), 50 ng IFNβ-firefly luciferase promoter, 2 ng FLAG-IKKε, and 5, 25, or 50 ng of empty vector or VP35-wt/K309R/K309G plasmids. Twenty-four hours after transfection, the cells were lysed and luciferase signal was measured using the Dual-Luciferase Reporter Assay System (Promega) with a Cytation 5 reader (Biotek). For the dsRNA-induction experiment, after 24 hours of plasmid transfection, HMW poly(I:C) (3.125 ug/mL) was transfected with Lipofectamine 2000 (Invitrogen). The cells were lysed at 16 hours after poly(I:C) transfection to measure luciferase. For both experiments, 30% of lysates were collected and boiled in 4X Laemmli sample buffer (Bio-Rad).

## Biotin-poly(I:C) binding

Biotin-labeled HMW poly(I:C), 500 ng, (InvivoGen) was allowed to bind streptavidin-agarose beads (Sigma) in NT2 buffer overnight at 4˚C on a rocking platform and washed seven times in NT2 buffer to remove any unbound poly(I:C). FLAG-peptide purified FLAG-VP35 was incubated with the biotin-poly(I:C) coated beads in 200 uL NT2 buffer for 4 hours at 4˚C on a rocking platform. After seven washes in NT2 buffer, the beads were boiled at 95˚C in 2X Laemmli sample buffer for 10 minutes.

## Minigenome assay

The monocistronic minigenome construct [6] previously modified by replacement of the chloramphenicol gene with the firefly luciferase gene [36] was kindly provided by Dr. Elke Mühlberger (BU). The plasmids pCEZ-NP, pCEZ-VP35, pCEZ-VP30, pCEZ-L, and pC-T7 [55] were kindly provided by Dr. Yoshihiro Kawaoka (UW). 293T cells were plated (50,000 cells/well) onto 24-well plates in 10% FBS 1X DMEM for 16 hours, and co-transfected with the following plasmids: EBOV minigenome (125 ng), pCEZ-VP30 (31.25 ng), pCEZ-NP (62.5 ng), pCEZ-L (500 ng), pC-T7 polymerase (125 ng), 25 or 100 ng of empty vector (pCAGGS) or pCAGGS-VP35 (wt, K309R, or K309G), and REN-Luc/pRL-TK plasmid (20 ng; Promega) expressing *Renilla* luciferase used as an internal control to normalize transfection efficiency. Fifty hours after transfection, the cells were lysed to measure luciferase signal as described above. A portion of the lysate was boiled at 95˚C for 10 minutes in 4X Laemmli buffer to evaluate protein expression via immunoblot.

## IFNβ ELISA

Supernatants from EBOV-infected A549 cells (MOI = 2.5 PFU/cell) were collected at 48 hpi to measure IFNβ using the VeriKine human IFN-β enzyme-linked immunosorbent assay (ELISA) kit (PBL Assay Science) according to the manufacturer's instructions. The limit of detection for the assay is 50 pg/ml.

## Western blots

Protein samples were run on 4–15% or 7.5% Mini-PROTEAN- or Criterion-TGX Precast Gels (Bio-Rad). The proteins were then transferred onto methanol-activated Immun-Blot PVDF membrane (Bio-Rad), and the membrane was blocked in 5% Carnation powdered skim milk (Nestle) in 1X TBS-T (blocking buffer) for 1 hour. Primary antibodies were prepared in 2% bovine serum albumin 1X TBS-T with 0.02% sodium azide to the appropriate dilution: anti-FLAG (Sigma) 1:2000, anti-HA (Sigma) 1:2000, anti-His (Sigma) 1:2000, anti-VP35 (6C5 Kerafast) 1:1000, anti-NP (provided by Dr. Basler, Mount Sinai) 1:1000, anti-VP30 (provided by Dr. Basler, Mount Sinai) 1:1000, anti-VP24 (Sino Biological) 1:1000, anti-VP40 (GeneTex) 1:1000, phosphorylated IRF3 S386 (Cell Signaling) 1:1000, total IRF3 (Immuno-Biological) 1:1000, phosphorylated TBK1 S172 (Epitomics) 1:1000, total TBK1 (Novus Biologicals) 1:1000, phosphorylated STAT1 Y701 (Cell Signaling) 1:1000, total STAT1 (BD Biosciences) 1:1000, anti-ubiquitin (Enzo) 1:1000, anti-TRIM6 (Sigma) 1:1000, anti-tubulin (Sigma) 1:2000, and anti-actin (Abcam) 1:2000. The next day, the blot was washed in 1X TBS-T prior to incubation with HRP-conjugated goat-anti-rabbit (GE Healthcare) or goat-anti-mouse (GE health care) for 1 hour. The blot was then washed and developed using Pierce ECL Western Blotting Substrate (Thermo Fisher) or SuperSignal West Femto Maximum Sensitivity Substrate (Thermo Scientific). For blot quantifications, the area under the curve (AUC) was measured for each band of interest using ImageJ [56].

## RNA extraction and quantitative PCR

Cells were lysed in Trizol (Thermo Fisher) or Tri-reagent (Zymo Research) and processed using the Direct-zol RNA kit (Zymo Research). For the standard qPCR reactions, cDNA was synthesized using the High-Capacity cDNA Reverse Transcription Kit (Applied Biosystems) following the manufacturer's instructions. The qPCR master mixes were prepared with *iTaq* Universal SYBR Green (Bio-Rad). The qPCR reactions were carried out using a CFX384 instrument (Bio-Rad). The relative mRNA expression levels were analyzed using CFX Manager software (Bio-Rad). The change in the threshold cycle (ΔCT) was calculated, with the 18S gene (human cells) or beta-actin (mouse cells) serving as the reference mRNA for normalization. The primers used to assess gene expression are listed in **S1 Table**.

For the strand-specific PCR, 200 ng (MEF) or 500 ng (A549 and VeroE6) of RNA was used for first-strand cDNA synthesis using the RevertAid First Strand cDNA Synthesis kit (Thermo Scientific). Three different first-strand reactions were performed with a vRNA, cRNA, or mRNA tagged primer (**S1 Table**). Prior to use in the qPCR reaction, the samples were diluted 1:5 in NF $H_2O$. To make the standards for quantification, minigenome plasmid DNA (vRNA and cRNA) or cDNA made from VeroE6 cells infected with EBOV (mRNAs) were cloned using the AccuPrime *Taq* DNA polymerase, high-fidelity kit (primers are listed in **S1 Table**). The forward (vRNA and mRNAs) or reverse (cRNA) primers included the T7 polymerase site. After running the PCR product on a 0.7% agarose 1X TAE gel at 90 V for 1 hour, the DNA was extracted and purified using the QIAquick Gel Extraction Kit (QIAGEN) and used as a template for in vitro transcription with the MEGAscript T7 Transcription Kit (Invitrogen). The RNA from the in vitro transcription was purified with the Direct-zol RNA kit with the on-column DNA digestion and quantified. In order to calculate the copy number with a standard curve, 1:10 dilutions ranging from $10^{10}$–$10^4$ of in vitro transcript cDNA was used in qPCR reactions (**S6A Fig**). To enumerate the copy number of each viral RNA species in the infected samples, the threshold cycle was plugged into the corresponding standard curve equation. To ensure the total RNA was similar between samples, standard qPCR for 18S and/or EBOV was run on cDNA generated with the high-capacity cDNA kit as described above. Limit of detection and specificity assays were also run to validate the assay (**S6B and S6C Fig**).

## Virus purification

Supernatants from a T75 flasks of VeroE6 cells infected at an MOI of 0.01 PFU/cell were collected at 144 hpi for sucrose-gradient purification. The 15 mL of supernatant was first clarified to remove cellular debris at 1000 g for 10 minutes before loading onto a 25% sucrose cushion. The virus was then pelleted using SW32 rotors spun at ~82,000 g for 2 hours in a Beckman-Coulter L90K ultracentrifuge. The pelleted virus was resuspended in 1X STE buffer (10 mM Tris, 1 mM EDTA, 0.1 M NaCl) and loaded onto a 20–60% sucrose gradient and spun at ~153,000 g for 1.5 hours in a SW41 rotor. The virus band was collected, diluted in 1X STE buffer, and pelleted again in SW32 rotors at ~82,000 g for 1 hour. The pellet was resuspended in 500 uL of 1X STE buffer. All spins were conducted at 4˚C. Aliquots of the purified virus were used for RNA or protein isolation and titration.

## ATPase assay

Peptide purified FLAG-VP35 was incubated in ATPase buffer (20 mM Tris-HCl pH 8.0, 1.5 mM $MgCl_2$, and 1.5 mM DTT in NF $H_2O$) with or without ATP (2.5 mM final concentration) at 37˚C for 30 minutes. Free phosphate was measured using the BIOMOL Green (Enzo) reagent for phosphate detection and read with the Cytation5 (620 nm). A standard curve was prepared with the provided phosphate standard according to the manufacturer's instructions.

## Statistics

All analyses were performed in Graphpad Prism (Version 7.04). Heat maps were also generated with Graphpad Prism. Statistical tests, measures of statistical significance, and replication information are specified in the respective figure legends. Repeated measures two-way ANOVA with Bonferroni's post-test was applied for two factor comparisons (kinetics experiments), one-way ANOVA with Tukey's post-test was used for comparing three or more groups, and a student's t-test for comparing two groups.

## Supporting information

**S1 Fig. rEBOV-VP35/K309 mutants are attenuated in additional cell types.** (**A**) A549 cells were infected with rEBOV-eGFP-VP35/wt, -K309R, or -K309G at an MOI of 2.5 PFU/cell for 24 hours. Limit of detection (LOD), 10 pfu/mL, is indicated. (**B-D**) Murine embryonic fibroblasts (MEFs) were infected with rEBOV-eGFP-VP35/wt, -K309R, or -K309G at an MOI of 10.0 PFU/cell. The titers are from 24 and 96 hpi, LOD 10pfu/mL (**B**), and EBOV RNA (**C**) and fluorescence microscopy images (**D**) are from 96 hpi. The titrations (**A and B**) and qRT-PCR (**C**) were done in triplicate. Analysis was done using a one-way ANOVA with Tukey's post-test for comparison between groups. P-value: $^{**}< 0.01$, $^{***}<0.001$, $^{****}<0.0001$; ns, not significant ($p>0.05$).
(TIF)

**S2 Fig. Ubiquitination of VP35 at K309 is not required for IFN-I antagonism.** (**A**) A549 cells were infected with rEBOV-eGFP-VP35/wt, -K309R, or -K309G at an MOI of 2.5 PFU/cell for 24 hours and RNA was collected for qRT-PCR. (**B**) Murine embryonic fibroblasts (MEFs) were infected with rEBOV-eGFP-VP35/wt, -K309R, or -K309G at an MOI of 10.0 PFU/cell for 48 hours and RNA was collected for qRT-PCR. The Ifnb and interferon stimulated genes (ISGs) RNA cycle threshold value was normalized to the 18S value. Analysis was done using a one-way ANOVA with Tukey's post-test for comparison between groups. P-value: $^{*}<0.05$, $^{**}<0.01$, $^{****}<0.0001$; ns, not significant ($p>0.05$).
(TIF)

**S3 Fig. Ebola virus is attenuated in TRIM6 knockout cells.** Wild-type (solid lines) and TRIM6 knockout (T6-KO) (dashed lines) A549 cells were infected with rEBOV-eGFP-VP35/wt, -K309R, or -K309G at multiplicity of infection (MOI) of 0.01 (**A**) and 2.5 (**B**) PFU/cell or mock treated (grey). The limit of detection (LOD) is 10 PFU/mL. The titrations were done in triplicate. The data analysis was done using a two-way ANOVA with Bonferroni's or Tukey's post-test for comparison between groups, respectively. P-value: $^{*}<0.05$, $^{***}<0.001$, $^{****}<0.0001$. Black, red and blue stars represent wt, K309G and K309R comparison, respectively, between wt and T6-KO cells. Non-significant differences are not indicated to limit crowding on the graph.
(TIF)

**S4 Fig. Ubiquitination of VP35 at K309 is required for viral transcription efficiency along the transcriptional gradient.** (**A**) VeroE6 cells were infected with rEBOV-eGFP-VP35/wt (black), -K309R (blue), or -K309G (red) at an MOI of 1.0 PFU/cell for 72 hours and RNA was collected for strand-specific qRT-PCR (two biological replicates run in triplicate). (**B**) Heat map representing the ratio of copy number relative to wt for each viral RNA species corresponding to the data presented in panel A. (**C**) Standard qPCR for viral RNA of VeroE6 cells at 24, 48, and 72 hpi (corresponding to samples used in Fig 3C, 3D and panel A). The EBOV RNA signal was normalized to the 18S cycle threshold value. (**D**) WT (black) or *Trim6*$^{-/-}$

(green) murine embryonic fibroblasts (MEFs) were infected with rEBOV-eGFP-VP35/wt at an MOI of 10.0 PFU/cell for 96 hours and RNA was collected for strand-specific qRT-PCR (triplicates). (**E**) Heat map representing the ratio of copy number relative to wt for each viral RNA species corresponding to the data presented in panel D. (**F**) The 8bp deletion (Δ8bp) in TRIM6 sequence of the TRIM6-KO mice used for MEF and bone marrow-derived dendritic cell generation. The highlighted sequence corresponds to the PAM, the sequence in red is the sgRNA target sequence, and the dashes represent the deleted nucleotides. The data analysis was done using a two-way ANOVA with Bonferroni's post-test for comparison between groups (**C**), one-way ANOVA with Tukey's post-test for comparison between groups (**A**) or a student's t-test (**D**). P-value: * $<0.05$, ** $<0.01$, *** $<0.001$, **** $<0.0001$; ns, not significant (p>0.05).
(TIF)

**S5 Fig. Mutation of VP35 at K309 does not alter VP35's ATPase activity.** FLAG-purified VP35 (wt, K309R, or K309G) was used in an ATPase activity assay. The concentration of free phosphate ($P_i$) was determined using a standard curve with the BIOMOL Green phosphate standard. A fraction of the completed reaction was boiled in 4X Laemmli sample buffer to compare the amount of VP35 added. The assay was completed in biological triplicate. The data analysis was done using a one-way ANOVA with Tukey's post-test for comparison between groups. No significant differences were identified.
(TIF)

**S6 Fig. Validation and standard curves for EBOV strand-specific qPCR.** (**A**) Representative standard curves for each viral RNA species tested, genomic RNA (vRNA), anti-genomic RNA (cRNA), and NP, VP35, VP40, VP24, and L transcripts (mRNA) run along with each strand-specific qPCR run. (**B**) Table of the limit of detection ($log_{10}$ copies) for each first-strand cDNA primer-in vitro transcript pair. (**C**) Graphical representation of how cycle threshold cut off values were determined for each first-strand primer/qPCR pair.
(TIF)

**S1 Table. Primer sequences.** A list of all used primer sequences. Sequences are listed 5'-to-3'.
(DOCX)

## Author Contributions

**Conceptualization:** Sarah van Tol, Alexander N. Freiberg, Ricardo Rajsbaum.

**Formal analysis:** Sarah van Tol.

**Funding acquisition:** Sarah van Tol, Alexander N. Freiberg, Ricardo Rajsbaum.

**Investigation:** Sarah van Tol, Birte Kalveram, Leopoldo Aguilera-Aguirre, Preeti Bharaj, Colm Atkins.

**Resources:** Philipp A. Ilinykh, Adam Ronk, Kai Huang, Adam Hage, Maria I. Giraldo, Maki Wakamiya, Maria Gonzalez-Orozco, Abbey N. Warren, Alexander Bukreyev.

**Supervision:** Alexander N. Freiberg, Ricardo Rajsbaum.

**Writing – original draft:** Sarah van Tol.

**Writing – review & editing:** Sarah van Tol, Birte Kalveram, Philipp A. Ilinykh, Adam Ronk, Kai Huang, Leopoldo Aguilera-Aguirre, Preeti Bharaj, Adam Hage, Colm Atkins, Maria I. Giraldo, Maki Wakamiya, Alexander Bukreyev, Alexander N. Freiberg, Ricardo Rajsbaum.

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
