## [Decision Letter · Decision Letter 0]

19 Jan 2022

Dear Dr Rajsbaum,

Thank you very much for submitting your manuscript "Ubiquitination of Ebola virus VP35 at lysine 309 regulates viral transcription and assembly" for consideration at PLOS Pathogens. As with all papers reviewed by the journal, your manuscript was reviewed by members of the editorial board and by several independent reviewers. In light of the reviews (below this email), we would like to invite the resubmission of a significantly-revised version that takes into account the reviewers' comments. Please note in particular the concerns of reviewer 2.

We cannot make any decision about publication until we have seen the revised manuscript and your response to the reviewers' comments. Your revised manuscript is also likely to be sent to reviewers for further evaluation.

Sincerely,

Jens H. Kuhn

Associate Editor

PLOS Pathogens

Christopher Basler

Section Editor

PLOS Pathogens

Kasturi Haldar

Editor-in-Chief

PLOS Pathogens

orcid.org/0000-0001-5065-158X

Michael Malim

Editor-in-Chief

PLOS Pathogens

orcid.org/0000-0002-7699-2064

Reviewer's Responses to Questions

**Part I - Summary**

Reviewer #1: In this manuscript, the authors continue their research on the role of Ebola virus VP35 ubiquitination for viral replication. By intensively investigations of two VP35 mutants, carrying substitutions at amino acids 309 to arginine or glycine residue, K309R or K309G, the authors could show by applying reporter gene assays, interaction studies and recombinant viruses, that ubiquitination of VP35 K309 is essential for interaction with the viral polymerase L in order to initiate efficient viral RNA synthesis. In contrast, lack of VP35 ubiquitination leads to stronger interaction with NP, allowing nucleocapsid assembly and packaging into progeny virions. All together, these results suggest ubiquitination of Ebola virus VP35 plays a key role in switching the two different functions of VP35 in transcription vs nucleocapsid assembly.

The manuscript is very well written and provides new evidence for the mechanistic role of VP35 ubiquitination during viral infection by modulating the two distinct function of VP35 in transcription and nucleocapsid assembly through ubiquitination of VP35 K309. The experiments are solidly executed and are very intensively controlled.

There are only minor issues to be addressed

Reviewer #2: This manuscript by van Tol et al., explores the role of Ebola VP35 ubiquitination. This work builds on a previous observation that Ebola VP35 is ubiquitinated on a specific basic residue, K309. The authors extend this initial observation by generating recombinant viruses with mutations at K309 (K209G and K309R) in an attempt to define the molecular mechanism. Much of the data in the manuscript support the original observation from the same group, Bharaj et al. 2017, with the major contribution, albeit not entirely surprising is the fact that mutations within the C-terminal RNA binding interferon inhibitory domain (RBD/IID) does not impact its interferon inhibitory function. This is consistent with multiple prior reports which suggested that K309 plays a minor role in the central basic patch region. Authors also generated recombinant viruses and this was important and has the potential to impact our knowledge. While the role of posttranslational modifications, including ubiquitination are likely to impact Ebola replication, the data in the manuscript overall is not rigorous. Often the error bars/statistical considerations are missing and the data/replicates are missing. Therefore, while the manuscript presents and important idea, the significance and the impact of the data lacks an ability to support the hypothesis. For example data in Figure 3 shows a log difference in panel A, but a black and white difference in panel C at MOI of 5 a dn 48 hours time. Panel C is using a normalization to actin. Using multiple Y-axis, the authors are either trying to convince themselves and the reader or enhance a modest effect. Unclear which of this is true. Finally, data in Figure 6 is intriguing. If the ubiquitination was truly the reason for why VP35 does not bind to Ebola viral L, then using WT VP35 and WT L1-505, one should see a disruption in TRIM6 KO cells. That data is missing. Again, in multiple instances, the authors do not provide rigorous data to support their ideas. For these reasons, there is promise, but does not deliver with respect to biology.

Reviewer #3: This is a thoroughly designed and executed study demonstrating the significance of ubiquitination of EBOV VP35 at lysine 309 in viral transcription and assembly. The authors generated recombinant K309R and K309G EBOV mutants to differentiate the role of ubiquitination and the basic property of the K309 residue in regulating virus infection. It appears that Lysine ubiquitination is not essential for VP35 to antagonize IFN-1 activity but was necessary for transcription and proper virus assembly. Several approaches have been used to drive the main conclusions. The data is also supported by rigorous statistical analysis.

The rescue of the mutant viruses and demonstration of their attenuating phenotype in vitro assays by itself is very promising for developing countermeasures. This is further enhanced by detailed molecular studies into the mechanism of action.

**Part II – Major Issues: Key Experiments Required for Acceptance**

Reviewer #1: There are only minor issues to be addressed

Reviewer #2: For example data in Figure 3 shows a log difference in panel A, but a black and white difference in panel C at MOI of 5 a dn 48 hours time. Panel C is using a normalization to actin. Using multiple Y-axis, the authors are either trying to convince themselves and the reader or enhance a modest effect. Unclear which of this is true. Finally, data in Figure 6 is intriguing. If the ubiquitination was truly the reason for why VP35 does not bind to Ebola viral L, then using WT VP35 and WT L1-505, one should see a disruption in TRIM6 KO cells. That data is missing.

Reviewer #3: I have one concern that needs clarification.

Fig. 6E. This is a very nice experiment; however interpretation of the results didn’t seem right due to the following reasons:

1) It appears that OTU co-expression with VP35 or VP35-K309R was sufficient to enhance VP35-NP interaction. This appears to be the case at both low and high expression of VP35. However, the corresponding densitometry analysis didn’t correlate with the Western blot data. Based on Western blot alone, it appears that there is some ubiquitin dependent component that was presumable destroyed by OTU that enhanced NP-VP35 interaction. This is in addition to the enhancement of NP-VP35 due to OTU mediated loss of ubiquitin modification at K309.

2) Can the authors demonstrate, as a control, the level of expression of the Ubiquitinated-VP35 species in the lysates of mock, OTU and 2A transfected cells. This will also check if there are any additional residues on VP35 (other than K309) that are ubiquitinated (like in Fig. 1c). It is possible that these additional residues are influencing VP35-NP interaction. However, it is understandable if the authors find such a demonstration to be technically challenging. But additional insights into this interesting observation should be provided.

In general the data just going by the Western blot is interesting but the interpretation was confusing and needs clarification to make the claim that NP-VP35 interaction is dependent only on K309 ubiquitination

**Part III – Minor Issues: Editorial and Data Presentation Modifications**

Reviewer #1: Minor comments.

1. Line 83: the abbreviation for antigenomic RNA, agRNA, is unusual. Conventionally, the terms and abbreviations for vRNA (genomic RNA) and cRNA (copy RNA, antigenome) are more widely used.

2. Line 175: To this reviewer, usage of A549 cells, a lung cell line, for Ebola virus infection seems unusual. There are more widely used IFN competent cells like Huh7 cells, a hepatoma cell line, liver cells are also infected during an authentic viral infections. Could the authors please comment on their rational to use the A549 cells?

3. Line 190 and line 193 Typo Ifnb. Must be IFNß. Isg must be ISG

4. Fig. 4A. To me, it is not clear why the authors also assed reporter gene activity at lower doses of VP35 and mutants. The ratio between VP35 and NP in the minigenome assay setting (Hoenen et al., 2004, Wenigenrath et al., 2010) is very critical for efficient transcirpiotnal activity. Altering this ratio by just transfecting a quarter of the usual dose of VP35 will of course affect transcriptional activity. However, this must not have anything to do with the VP35 Ub mutations at all.

5. Fig. 6C. Expression control of L is missing.

6. Fig. 6F. A quantification of Western Blots is missing. While the authors state that interaction of mutants with VP30 is not impaired (line 346), the blot shows another picture. Also with respect to Fig 7A, where K309G infection shows reduced VP30 incorporatio, it is highly recommended to include here a statistical analysis of more than 1 Blot to interprete these results more precisely.

7. Fig. 8. It is highly recommended to replace the reddish, pinkish colours with different types to improve clarity of the model. Also the particles that are “produced” are really hard to recognize instinctively.

Reviewer #2: statistics/repeats, use of acceptable Y-axis and describe rationale when unusual nationalizations are done.

Reviewer #3: No issues in this section

PLOS authors have the option to publish the peer review history of their article (what does this mean?). If published, this will include your full peer review and any attached files.

Reviewer #1: No

Reviewer #2: No

Reviewer #3: No
---

## [Decision Letter · Decision Letter 1]

18 Apr 2022

Dear Dr Rajsbaum,

We are pleased to inform you that your manuscript 'Ubiquitination of Ebola virus VP35 at lysine 309 regulates viral transcription and assembly' has been provisionally accepted for publication in PLOS Pathogens.

Best regards,

Jens H. Kuhn

Associate Editor

PLOS Pathogens

Christopher Basler

Section Editor

PLOS Pathogens

Kasturi Haldar

Editor-in-Chief

PLOS Pathogens

orcid.org/0000-0001-5065-158X

Michael Malim

Editor-in-Chief

PLOS Pathogens

orcid.org/0000-0002-7699-2064

Reviewer Comments (if any, and for reference):

Reviewer's Responses to Questions

**Part I - Summary**

Reviewer #1: The authors present a thouroughly revised manuscript where they also included important new experiments that support there hypothesis in the role of VP35 K309 Ub. They also revised there statistical analysis.

**Part II – Major Issues: Key Experiments Required for Acceptance**

Reviewer #1: n.a.

**Part III – Minor Issues: Editorial and Data Presentation Modifications**

Reviewer #1: All minor issues were covered.

PLOS authors have the option to publish the peer review history of their article (what does this mean?). If published, this will include your full peer review and any attached files.

Reviewer #1: No

---

## [Editor Report · Acceptance letter]

5 May 2022

Dear Dr Rajsbaum,

We are delighted to inform you that your manuscript, "Ubiquitination of Ebola virus VP35 at lysine 309 regulates viral transcription and assembly," has been formally accepted for publication in PLOS Pathogens.

Best regards,

Kasturi Haldar

Editor-in-Chief

PLOS Pathogens

orcid.org/0000-0001-5065-158X

Michael Malim

Editor-in-Chief

PLOS Pathogens

orcid.org/0000-0002-7699-2064